# On Memory Construction and Retrieval for Personalized Conversational Agents

**Zhuoshi Pan**[1][†] **Qianhui Wu**[2][‡] **Huiqiang Jiang**[2], **Xufang Luo**[2], **Hao Cheng**[2],
**Dongsheng Li**[2], **Yuqing Yang**[2], **Chin-Yew Lin**[2], **H. Vicky Zhao**[1][‡] **Lili Qiu**[2], **Jianfeng Gao**[2]

[1] Tsinghua University, [2] Microsoft Corporation

## Abstract

To deliver coherent and personalized experiences in long-term conversations, existing approaches typically perform retrieval augmented response generation by constructing memory banks from conversation history at either the turn-level, session-level, or through summarization techniques. In this paper, we present two key findings: (1) The granularity of memory unit matters: Turn-level, session-level, and summarization-based methods each exhibit limitations in both memory retrieval accuracy and the semantic quality of the retrieved content. (2) Prompt compression methods, such as *LLMLingua-2*, can effectively serve as a denoising mechanism, enhancing memory retrieval accuracy across different granularities.

Building on these insights, we propose **SECOM**, a method that constructs the memory bank at segment level by introducing a conversation **SE**gmentation model that partitions long-term conversations into topically coherent segments, while applying **COM**pression based denoising on memory units to enhance memory retrieval. Experimental results show that SECOM exhibits a significant performance advantage over baselines on long-term conversation benchmarks *LOCOMO* and *Long-MT-Bench+*. Additionally, the proposed conversation segmentation method demonstrates superior performance on dialogue segmentation datasets such as *DialSeg711*, *TIAGE*, and *SuperDialSeg*.

## 1 Introduction

Large language models (LLMs) have developed rapidly in recent years and have been widely used in conversational agents. In contrast to traditional dialogue systems, which typically focus on short conversations within specific domains (Dinan et al., 2019), LLM-powered conversational agents engage in significantly more interaction turns across a broader range of topics in open-domain conversations (Kim et al., 2023; Zhou et al., 2023). Such long-term, open-domain conversations over multiple sessions present significant challenges, as they require the system to retain past events and user preferences to deliver coherent and personalized responses (Chen et al., 2024).

Some methods maintain context by concatenating all historical utterances or their summarized versions (LangChain Team, 2023a; Wang et al., 2023). However, these strategies can result in excessively long contexts that include irrelevant information, which may not be relevant to the user's current request. As noted by Maharana et al. (2024), LLMs struggle with understanding lengthy conversations and grasping long-range temporal and causal dynamics, particularly when the dialogues contain irrelevant information (Jiang et al., 2023c). Some other works focus on retrieving query-related conversation history to enhance response generation (Yuan et al., 2023; Alonso et al., 2024; Kim et al., 2024; Maharana et al., 2024). These approaches typically construct a memory bank from the conversation history at either the *turn-level* (Yuan et al., 2023) or *session-level* (Wang et al., 2023). Chen et al. (2024), Li et al. (2024) and Zhong et al. (2024) further leverage *summarization* techniques to build memory units, which are then retrieved as context for response generation.

---

[†]Work during internship at Microsoft.
[‡]Corresponding author.
[§]Project page: https://aka.ms/secom

Building on these works, a key question arises: Which level of memory granularity—turn-level, session-level, or their summarized forms—yields the highest effectiveness? Moreover, is there a novel memory structure that could outperform these three formats?

In this paper, we first systematically investigate the impact of different memory granularities on conversational agents within the paradigm of retrieval augmented response generation (Lewis et al., 2020; Ye et al., 2024). Our findings indicate that turn-level, session-level, and summarization-based methods all exhibit limitations in terms of the accuracy of the retrieval module as well as the semantics of the retrieved content, which ultimately lead to sub-optimal responses, as depicted in Figure 1, Figure 2, and Table 1.

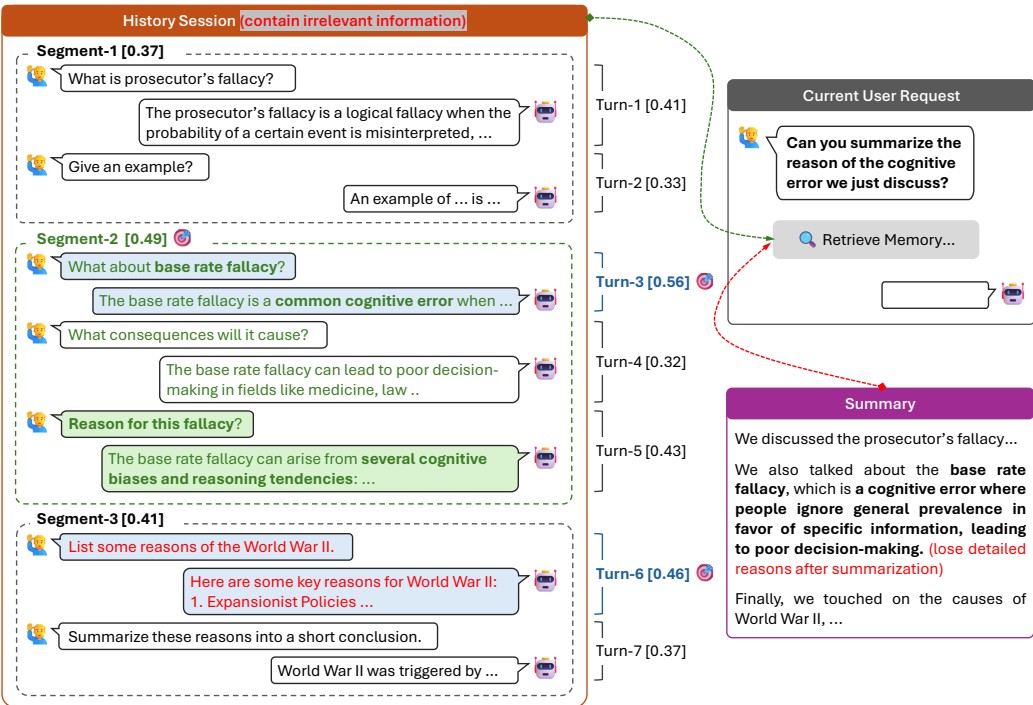

Figure 1: Illustration of retrieval augmented response generation with different memory granularities. *Turn-level memory* is too fine-grained, leading to fragmentary and incomplete context. *Session-level memory* is too coarse-grained, containing too much irrelevant information. *Summary based methods* suffer from information loss that occurs during summarization. *Ours (segment-level memory)* can better capture topically coherent units in long conversations, striking a balance between including more relevant, coherent information while excluding irrelevant content. Bullseye ⊙ indicates the retrieved memory units at turn level or segment level under the same context budget. [0.xx]: similarity between target query and history content. Turn-level retrieval errors: false negative , false positive .

Specifically, users often interact with agents over multiple turns to achieve their goals, causing relevant information to be dispersed across multiple interactions. This dispersion can pose a great challenge to the retrieval of turn-level memory units as some of the history conversation turns may not explicitly contain or relate to keywords mentioned in the current request (*e.g.*, Turn-5 in Figure 1). As a result, the retrieved contexts (*e.g.*, Turn-3 and Turn-6 in Figure 1) can be fragmentary and fail to encompass the complete request-related information flow, leading to responses that may lack coherence or omit essential information. On the other hand, a single conversation session may cover multiple topics, especially when users do not initiate a new chat session upon switching topics. Therefore, constructing memory units at the session level risks including irrelevant content (*e.g.*, definition of the prosecutor's fallacy and reasons of World War II in Figure 1). Such extraneous content in the session-level memory unit may not only distract the retrieval module but also disrupt the language model's comprehension of the context, causing the agent to produce responses that are off-topic or include unnecessary details.

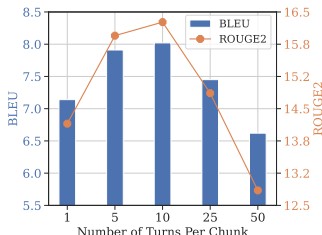 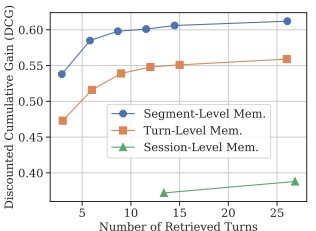 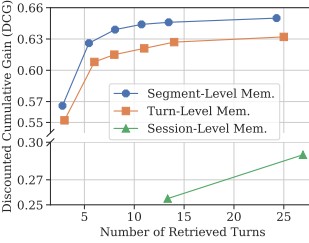

(a) Response quality as a function of chunk size, given a total budget of 50 turns to retrieve as context.

(b) Retrieval DCG obtained with different memory granularities using BM25 based retriever.

(c) Retrieval DCG obtained with different memory granularities using MPNet based retriever.

Figure 2: The impact of memory granularity on the response quality (a) and retrieval accuracy (b, c).

Long conversations are naturally composed of coherent discourse units. To capture this structure, we introduce a conversation segmentation model that partitions long-term conversations into topically coherent segments, constructing the memory bank at the segment level. During response generation, we directly concatenate the retrieved segment-level memory units as the context as in Yuan et al. (2023); Kim et al. (2024), bypassing summarization to avoid the information loss that often occurs when converting dialogues into summaries (Maharana et al., 2024).

Furthermore, inspired by the notion that natural language tends to be inherently redundant (Shannon, 1951; Jiang et al., 2023b; Pan et al., 2024), we hypothesize that such redundancy can act as noise for retrieval systems, complicating the extraction of key information (Grangier et al., 2003; Ma et al., 2021). Therefore, we propose removing such redundancy from memory units prior to retrieval by leveraging prompt compression methods such as LLMLingua-2 (Pan et al., 2024). Figure 3 shows the results obtained with a BM25 based retriever and an MPNet based retriever (Song et al., 2020) on *Long-MT-Bench+*. As demonstrated in Figure 3a and Figure 3b, LLMLingua-2 consistently improves retrieval recall given different retrieval budgets $K$ (*i.e.*, the number of retrieved segments) when the compression rate exceeds 50%. Figure 3c further illustrates that, after denoising, similarity between the query and relevant segments increases, while the similarity with irrelevant segments decreases.

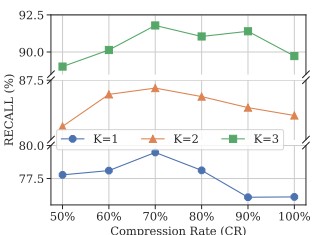 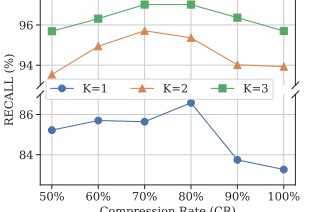 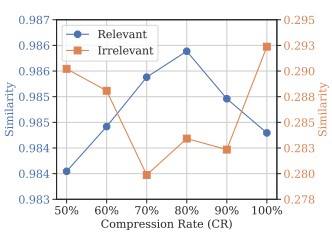

(a) Retrieval recall v.s. compression rate: $\frac{\text{\# tokens after compression}}{\text{\# tokens before compression}}$.
K: number of retrieved segments.
Retriever: BM25

(b) Retrieval recall v.s. compression rate: $\frac{\text{\# tokens after compression}}{\text{\# tokens before compression}}$.
K: number of retrieved segments.
Retriever: MPNet

(c) Similarity between the query and different dialogue segments. Blue: relevant segments. Orange: irrelevant segments. Retriever: MPNet

Figure 3: Prompt compression method (LLMLingua-2) can serve as an effective denoising technique to enhance the memory retrieval system by: (a) improving the retrieval recall with varying context budget $K$; (b) benefiting the retrieval system by increasing the similarity between the query and relevant segments while decreasing the similarity with irrelevant ones.

Our contributions can be summarized as follows:

- We systematically investigate the effects of memory granularity on retrieval augmented response generation in conversational agents. Our findings reveal that turn-level, session-level, and summarization-based approaches each face challenges in ensuring precise retrieval and providing a complete, relevant, and coherent context for generating accurate responses.

- We contend that the inherent redundancy in natural language can act as noise for retrieval systems. We demonstrate that prompt compression technique, LLMLingua-2, can serve as an effective denoising method to enhance memory retrieval performance.
- We present SECOM, a system that constructs memory bank at segment level by introducing a conversation SEgmentation model, while applying COMpression based denoising on memory units to enhance memory retrieval. The experimental results show that SECOM outperforms baselines on two long-term conversation benchmark LOCOMO and Long-MT-Bench+. Further analysis and ablation studies confirm the contributions of the segment-level memory units and the compression-based denoising technique within our framework.

## 2 SECOM

### 2.1 PRELIMINARY

Let $\mathcal{H} = \{c_i\}_{i=1}^{C}$ represent the available conversation history between a user and an agent, which consists of $C$ sessions. $c_i = \{t_j\}_{j=1}^{T_i}$ denotes the $i$-th session that is composed of $T_i$ sequential user-agent interaction turns, with each turn $t_j = (u_j, r_j)$ consisting of a user request $u_j$ and the corresponding response from the agent $r_j$. Denote the base retrieval system as $f_R$ and the response generation model as $f_{\text{LLM}}$. The research framework here can be defined as: (1) *Memory construction*: construct a memory bank $\mathcal{M}$ using conversation history $\mathcal{H}$; For a turn-level memory bank, each memory unit $m \in \mathcal{M}$ corresponds to an interaction turn $t$, with $|\mathcal{M}| = \sum_{i=1}^{C} T_i$. For a session-level memory bank, each memory unit $m$ corresponds to a session $c$, with $|\mathcal{M}| = C$. (2) *Memory retrieval*: given a target user request $u^*$ and context budget $N$, retrieve $N$ memory units $\{m_n \in \mathcal{M}\}_{n=1}^{N} \leftarrow f_R(u^*, \mathcal{M}, N)$ that are relevant to user request $u^*$; (3) *Response generation*: take the retrieved $N$ memory units in time order as the context and query the response generation model for response $r^* = f_{\text{LLM}}(u^*, \{m_n\}_{n=1}^{N})$.

In the remainder of this section, we first elaborate on the proposed conversation segmentation model that splits each session $c_i$ into $K_i$ topical segments $\{s_k\}_{k=1}^{K_i}$ in Section 2.2, with which we construct a segment-level memory bank with each memory unit $m$ corresponding to a segment $s$ and $|\mathcal{M}| = \sum_{i=1}^{C} K_i$. In Section 2.3, we describe how to denoise memory units to enhance the accuracy of memory retrieval.

### 2.2 CONVERSATION SEGMENTATION

**Zero-shot Segmentation**  Given a conversation session $c$, the conversation segmentation model $f_{\mathcal{I}}$ aims to identify *a set of segment indices* $\mathcal{I} = \{(p_k, q_k)\}_{k=1}^{K}$, where $K$ denotes the total number of segments within the session $c$, $p_k$ and $q_k$ represent the indexes of the first and last interaction turns for the $k$-th segment $s_k$, with $p_k \leq q_k, p_{k+1} = q_k + 1$. This can be formulated as:

$$f_{\mathcal{I}}(c) = \{s_k\}_{k=1}^{K}, \text{where } s_k = \{t_{p_k}, t_{p_k+1}, ..., t_{q_k}\} \tag{1}$$

However, building a segmentation model for open-domain conversation is challenging, primarily due to the difficulty of acquiring large amounts of annotated data. As noted by Jiang et al. (2023d), the ambiguous nature of segmentation points complicates data collection, making the task difficult even for human annotators. Consequently, we employ GPT-4 as the conversation segmentation model $f_{\mathcal{I}}$ to leverage its powerful text understanding ability across various domains. To provide clearer context and facilitate reasoning, we enhance session data $c$ by adding turn indices and role identifiers to each interaction $t_j$ as: "Turn $j$: \n[user]: $u_j$ \n[agent]: $r_j$". We empirically demonstrate that segmentation can also be accomplished with more lightweight models, such as *Mistral-7B* and even *RoBERTa* scale models, making our approach applicable in resource-constrained environments. Figure 6 in Appendix A.1 presents the detailed instruction used for zero-shot conversation segmentation here.

**Segmentation with Reflection on Limited Annotated Data**  When a small amount of conversation data with segment annotations is available, we leverage this annotated data to inject segmentation knowledge into LLMs and better align the LLM-based segmentation model with human preferences. Inspired by the prefix-tuning technique (Li & Liang, 2021) and reflection mechanism (Shinn et al., 2023; Renze & Guven, 2024), we treat the segmentation prompt as the "prefix" and iteratively optimize it through LLM self-reflection, ultimately obtaining a segmentation guidance $G$.

Specifically, in each iteration, we first apply our segmentation model in a zero-shot manner to a batch of conversation data and select the "hard examples", *i.e.,* the top $K$ sessions with the most significant segmentation errors based on the WindowDiff metric (Pevzner & Hearst, 2002). The LLM-based segmentation model is then instructed to reflect on its mistakes given the ground-truth segmentation annotations and update the segmentation guidance $\boldsymbol{G}$. This process mirrors Stochastic Gradient Descent (SGD) optimization, *i.e.,* $\boldsymbol{G}_{m+1} = \boldsymbol{G}_m - \eta \nabla \mathcal{L}(\boldsymbol{G}_m)$, where $\nabla \mathcal{L}(\boldsymbol{G}_m)$ denotes the gradient of segmentation loss, which we assume is estimated implicitly by the LLM itself and is used to adjust the next segmentation guidance $\boldsymbol{G}_{m+1}$. Figure 7 shows the self-reflection prompt and Figure 8 illustrates the final prompt with the learned rubric for segmentation.

## 2.3 COMPRESSION BASED MEMORY DENOISING

Given a target user request $u^*$ and context budget $N$, the memory retrieval system $f_R$ retrieves $N$ memory units $\{\boldsymbol{m}_n \in \mathcal{M}\}_{n=1}^N$ from the memory bank $\mathcal{M}$ as the context in response to the user request $u^*$. With the consideration that the inherent redundancy in natural language can act as noise for the retrieval system (Grangier et al., 2003; Ma et al., 2021), we denoise memory units by removing such redundancy via a prompt compression model $f_{Comp}$ before retrieval:

$$\{\boldsymbol{m}_n \in \mathcal{M}\}_{n=1}^N \leftarrow f_R(u^*, f_{Comp}(\mathcal{M}), N). \tag{2}$$

Specifically, we use LLMLingua-2 (Pan et al., 2024) as the denoising function $f_{Comp}$ here.

## 3 EXPERIMENTS

**Implementation Details** We use `GPT-35-Turbo` for response generation in our main experiment. We also adopt `Mistral-7B-Instruct-v0.3`[*] (Jiang et al., 2023a) for robustness evaluation across different LLMs. We employ zero-shot segmentation for QA benchmarks and further incorporate the reflection mechanism for segmentation benchmarks to leverage the available annotated data. To make our method applicable in resource-constrained environments, we conduct additional experiments by using `Mistral-7B-Instruct-v0.3` and a `RoBERTa` based model fine-tuned on SuperDialseg (Jiang et al., 2023d). Details for the conversation segmentation such as the prompt and hyper-parameters are described in Appendix A.1. We use `LLMLingua-2` (Pan et al., 2024) with a compression rate of 75% and `xlm-roberta-large` (Conneau et al., 2020) as the base model to denoise memory units. Following Alonso et al. (2024), we apply MPNet (`multi-qa-mpnet-base-dot-v1`) (Song et al., 2020) with FAISS (Johnson et al., 2019) and BM25 (Amati, 2009) for memory retrieval.

**Datasets & Evaluation Metrics** We evaluate SECOM and other baseline methods for long-term conversations on the following benchmarks:

(i) *LOCOMO* (Maharana et al., 2024), which is the longest conversation dataset to date, with an average of 300 turns with 9K tokens per sample. For the test set, we prompt GPT-4 to generate QA pairs for each session as in Alonso et al. (2024). We also conduct evaluation on the recently released official QA pairs in Appendix A.5.

(ii) *Long-MT-Bench+*, which is reconstructed from *MT-Bench+* (Lu et al., 2023), where human experts are invited to expand the original questions and create long-range questions as test user requests. Since each conversation only contains an average of 13.3 dialogue turns, following Yuan et al. (2023), we merge five consecutive sessions into one long-term conversation. We also use these human-written questions as few-shot examples to prompt GPT-4 to generate a long-range test question for each dialogue topic as the test set. More details such as the statistics of the constructed *Long-MT-Bench+* are listed in Appendix A.7.

For evaluation metrics, we use the conventional *BLEU* (Papineni et al., 2002), *ROUGE* (Lin, 2004), and *BERTScore* (Zhang et al., 2020) for basic evaluation. Inspired by (Pan et al., 2023), we employ *GPT4Score* for more accurate evaluation, where `GPT-4-0125` is prompted to assign an integer rating from 0 (poor) to 100 (excellent). We also perform *pairwise comparisons* by instructing GPT-4 to determine the superior response. The evaluation prompts are detailed in Figure 12 of Appendix A.4. Human evaluation is also conducted, with results summarized in Table 10 in Appendix A.10.

---

[*]`https://huggingface.co/mistralai/Mistral-7B-Instruct-v0.3`

Table 1: Performance comparison on *LOCOMO* and *Long-MT-Bench+*. The context budget for memory retrieval is set to 4k tokens ($\sim$ 5 sessions, 10 segments, or 55 turns) on *LOCOMO* and 1k tokens ($\sim$ 1 segments, 3 turns) on *Long-MT-Bench+*.

| Methods | QA Performance | | | | | | Context Length | |
|---|---|---|---|---|---|---|---|---|
| | GPT4Score | BLEU | Rouge1 | Rouge2 | RougeL | BERTScore | # Turns | # Tokens |
| *LOCOMO* | | | | | | | | |
| Zero History | 24.86 | 1.94 | 17.36 | 3.72 | 13.24 | 85.83 | 0.00 | 0 |
| Full History | 54.15 | 6.26 | 27.20 | 12.07 | 22.39 | 88.06 | 210.34 | 13,330 |
| Turn-Level (BM25) | 65.58 | 7.05 | 29.12 | 13.87 | 24.21 | 88.44 | 49.82 | 3,657 |
| Turn-Level (MPNet) | 57.99 | 6.07 | 26.61 | 11.38 | 21.60 | 88.01 | 54.77 | 3,288 |
| Session-Level (BM25) | 63.16 | 7.45 | 29.29 | 14.24 | 24.29 | 88.33 | 55.88 | 3,619 |
| Session-Level (MPNet) | 51.18 | 5.22 | 24.23 | 9.33 | 19.51 | 87.45 | 53.88 | 3,471 |
| SumMem | 53.87 | 2.87 | 20.71 | 6.66 | 16.25 | 86.88 | - | 4,108 |
| RecurSum | 56.25 | 2.22 | 20.04 | 8.36 | 16.25 | 86.47 | - | 400 |
| ConditionMem | 65.92 | 3.41 | 22.28 | 7.86 | 17.54 | 87.23 | - | 3,563 |
| MemoChat | 65.10 | 6.76 | 28.54 | 12.93 | 23.65 | 88.13 | - | 1,159 |
| SECOM (BM25, GPT4-Seg) | **71.57** | **8.07** | **31.40** | **16.30** | **26.55** | **88.88** | 55.52 | 3,731 |
| SECOM (MPNet, GPT4-Seg) | 69.33 | 7.19 | 29.58 | 13.74 | 24.38 | 88.60 | 55.51 | 3,716 |
| SECOM (MPNet, Mistral-7B-Seg) | 66.37 | 6.95 | 28.86 | 13.21 | 23.96 | 88.27 | 55.80 | 3,720 |
| SECOM (MPNet, RoBERTa-Seg) | 61.84 | 6.41 | 27.51 | 12.27 | 23.06 | 88.08 | 56.32 | 3,767 |
| *Long-MT-Bench+* | | | | | | | | |
| Zero History | 49.73 | 4.38 | 18.69 | 6.98 | 13.94 | 84.22 | 0.00 | 0 |
| Full History | 63.85 | 7.51 | 26.54 | 12.87 | 20.76 | 85.90 | 65.45 | 19,287 |
| Turn-Level (BM25) | 82.85 | 11.52 | 32.84 | 17.86 | 26.03 | 87.03 | 3.00 | 1,047 |
| Turn-Level (MPNet) | 84.91 | 12.09 | 34.31 | 19.08 | **27.82** | 86.49 | 3.00 | 909 |
| Session-Level (BM25) | 81.27 | 11.85 | 32.87 | 17.83 | 26.82 | 87.32 | 13.35 | 4,118 |
| Session-Level (MPNet) | 73.38 | 8.89 | 29.34 | 14.30 | 22.79 | 86.61 | 13.43 | 3,680 |
| SumMem | 63.42 | 7.84 | 25.48 | 10.61 | 18.66 | 85.70 | - | 1,651 |
| RecurSum | 62.96 | 7.17 | 22.53 | 9.42 | 16.97 | 84.90 | - | 567 |
| ConditionMem | 63.55 | 7.82 | 26.18 | 11.40 | 19.56 | 86.10 | - | 1,085 |
| MemoChat | 85.14 | 12.66 | 33.84 | 19.01 | 26.87 | 87.21 | - | 1,615 |
| SECOM (BM25, GPT4-Seg) | 86.67 | 12.74 | 33.82 | 18.72 | 26.87 | 87.37 | 2.87 | 906 |
| SECOM (MPNet, GPT4-Seg) | **88.81** | **13.80** | **34.63** | **19.21** | 27.64 | **87.72** | 2.77 | 820 |
| SECOM (MPNet, Mistral-7B-Seg) | 86.32 | 12.41 | 34.37 | 19.01 | 26.94 | 87.43 | 2.85 | 834 |
| SECOM (MPNet, RoBERTa-Seg) | 81.52 | 11.27 | 32.66 | 16.23 | 25.51 | 86.63 | 2.96 | 841 |

**Baselines** We evaluate our method against four intuitive approaches and four state-of-the-art models. As Figure 3 indicates, the compression-based memory denoising mechanism can benefit memory retrieval, in the main results, we directly compare our method to the denoising-enhanced turn-level and session-level baselines. (1) *Turn-Level*, which constructs the memory bank by treating each user-agent interaction as a distinct memory unit. (2) *Session-Level*, which uses each entire conversation session as a memory unit. (3) *Zero History*, which generates responses without incorporating any conversation history, operating in a zero-shot manner. (4) *Full History*, which concatenates all prior conversation history as the context for response generation. (5) *SumMem* (LangChain Team, 2023c), which dynamically generates summaries of past dialogues relevant to the target user request, and uses these summaries as context for response generation. (6) *RecurSum* (Wang et al., 2023), which recursively updates summary using current session and previous summaries, and takes the updated summary of current session as the context. (7) *ConditionMem* (Yuan et al., 2023), which generates summaries and knowledge for each dialogue turn, then retrieves the most relevant summary, knowledge, and raw conversation turn as the context in response to a new user request. (8) *MemoChat* (Lu et al., 2023), which operates memories at segment level, but focuses on tuning LLMs for both memory construction and retrieval.

**Main Results** As shown in Table 1 and Figure 4, SECOM *outperforms all baseline approaches*, exhibiting a significant performance advantage, particularly on the long-conversation benchmark LOCOMO. Interestingly, there is a significant performance disparity in Turn-Level and Session-Level methods when using different retrieval models. For instance, switching from the MPNet-based retriever to the BM25-based retriever results in performance improvements up to 11.98 and 7.89 points in terms of GPT4Score on LOCOMO and Long-MT-Bench+, respectively. In contrast, SECOM

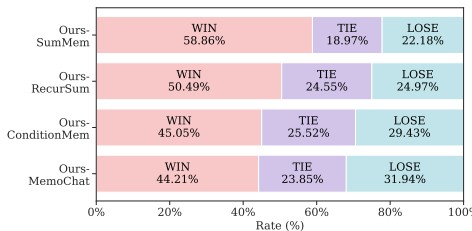 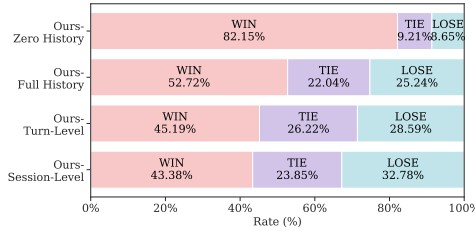

(a) SECOM *v.s.* state-of-the-art methods     (b) SECOM (segment-level) *v.s.* other granularities

Figure 4: GPT-4 based pairwise performance comparison on LOCOMO with BM25 based retriever.

*demonstrates greater robustness in terms of the deployed retrieval system.* We attribute this to the following reason: As discussed in Section 1, turn-level memory units are often fragmented and may not explicitly include or relate to keywords mentioned in the target user request. On the other hand, session-level memory units contain a large amount of irrelevant information. Both of these scenarios make the retrieval performance sensitive to the capability of the deployed retrieval system. However, topical segments in SECOM can strike a balance between including more relevant, coherent information while excluding irrelevant content, thus leading to more robust and superior retrieval performance. Table 1 and Figure 4 also reveal that *summary based methods, such as SumMem and RecurSum fall behind turn-level or session-level baselines.* Our case study, Figure 15 and 16 in Appendix A.6, suggests that this is likely due to the loss of crucial details during the process of converting dialogues into summaries (Maharana et al., 2024), which are essential for accurate question answering. Furthermore, Table 1 shows that SECOM maintains the advantage over baseline methods when switching the segmentation model from GPT-4 to Mistral-7B. Notably, even with a RoBERTa-based segmentation model, SECOM retains competitive performance compared to other granularity-based baselines.

**Ablation Study on Granularity of Memory Units**    Figure 2b, Figure 2c, and Table 3 have clearly demonstrated the superiority of segment-level memory over turn-level and session-level memory in terms of both retrieval accuracy and end-to-end QA performance. Figure 5a and Figure 5b further compare QA performance across different memory granularities under varying context budgets. Compression-based memory unit denoising was applied in all experiments here to isolate the end-to-end impact of memory granularity on performance. The results show that segment-level memory consistently outperforms both turn-level and session-level memory across a range of context budgets, reaffirming its superiority. Figures 13 and 14 in Appendix A.6 provide detailed case studies.

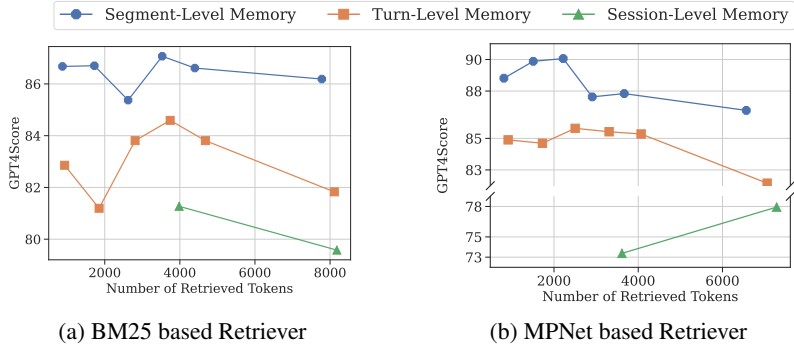

(a) BM25 based Retriever        (b) MPNet based Retriever

Figure 5: Performance comparison of different memory granularities with various context budget on *Long-MT-Bench+*.

**Ablation Study on Compression based Memory Denoising**    As shown in Table 2, removing the proposed compression based memory denoising mechanism will result in a performance drop up to 9.46 points of GPT4Score on LOCOMO, highlighting the critical role of this denoising

Table 2: Ablation study on compression-based memory denoising with a compression rate of 75% using the MPNet based retriever.

| Methods | LOCOMO | | | | Long-MT-Bench+ | | | |
|---------|--------|------|--------|-----------|----------------|------|--------|-----------|
| | GPT4Score | BLEU | Rouge2 | BERTScore | GPT4Score | BLEU | Rouge2 | BERTScore |
| SECOM | **69.33** | **7.19** | **13.74** | **88.60** | **88.81** | **13.80** | **19.21** | **87.72** |
| − Denoise | 59.87 | 6.49 | 12.11 | 88.16 | 87.51 | 12.94 | 18.73 | 87.44 |

mechanism: by effectively improving the retrieval system (Figure 3b), it significantly enhances the overall effectiveness of the system.

**Mistral-7B Powered Response Generation**    Table 3 presents the results of SECOM and baselines using `Mistral-7B-Instruct-v0.3`[†] (Jiang et al., 2023a) as the response generator. Our method demonstrates a significant performance gain over other baselines, showcasing its good generalization ability across different LLM-powered conversation agents. Interestingly, although the Mistral-7B here features a 32K context window capable of accommodating the entire conversation history, in other words, it is able to include and comprehend the entire conversation history without truncation, the performance of the "Full History" approach still falls short compared to SECOM. This highlights the effectiveness of our memory construction and retrieval mechanisms, which prioritize relevant context and reduce noise, leading to more accurate and contextually appropriate responses.

Table 3: Performance comparison on *Long-MT-Bench+* using `Mistral-7B-Instruct-v0.3`. Other settings are the same as Table 1.

| Methods | QA Performance | | | | | | Context Length | |
|---------|-----------|------|--------|--------|--------|-----------|---------|----------|
| | GPT4Score | BLEU | Rouge1 | Rouge2 | RougeL | BERTScore | # Turns | # Tokens |
| Full History | 78.73 | 10.25 | 29.43 | 14.32 | 23.37 | 86.77 | 65.45 | 19,287 |
| *BM25 Based Retriever* | | | | | | | | |
| Turn-Level | 83.14 | 13.60 | 33.28 | 19.11 | 27.32 | 87.52 | 3.00 | 1,047 |
| Session-Level | 81.03 | 12.49 | 32.39 | 17.11 | 25.66 | 87.21 | 13.35 | 4,118 |
| SECOM | **89.43** | **15.06** | **35.77** | **21.35** | **29.50** | **87.89** | 2.87 | 906 |
| *MPNet Based Retriever* | | | | | | | | |
| Turn-Level | 85.61 | 12.78 | 35.06 | 19.61 | 28.51 | 87.77 | 3.00 | 909 |
| Session-Level | 75.29 | 9.14 | 28.65 | 13.91 | 22.52 | 86.51 | 13.43 | 3,680 |
| SECOM | **90.58** | **15.80** | **36.14** | **21.49** | **29.94** | **88.07** | 2.77 | 820 |

**Evaluation of Conversation Segmentation Model**    To evaluate the conversation segmentation module described in Section 2.2 independently, we use three widely used dialogue segmentation datasets: DialSeg711 (Xu et al., 2021), TIAGE (Xie et al., 2021), and SuperDialSeg (Jiang et al., 2023d). In addition to the unsupervised (zero-shot) setting, we also assess performance in a transfer learning setting, where baseline models are trained on the full training set of the source dataset, while our model learns the segmentation rubric through LLM reflection on the top 100 most challenging examples. We evaluate transfer learning only using SuperDialSeg and TIAGE as the source datasets since DialSeg711 lacks a training set. For evaluation metrics, following Jiang et al. (2023d), we use the F1 score, $P_k$ (Beeferman et al., 1999), Window Diff (WD) (Pevzner & Hearst, 2002) and the segment score[‡]:

$$Score = \frac{2 * F1 + (1 - P_k) + (1 - WD)}{4}. \tag{3}$$

Table 4 presents the results, showing that our segmentation model consistently outperforms baselines in the unsupervised setting. In the transfer learning setting, despite the segmentation rubric being learned from LLM reflection on only 100 examples from the source dataset, it generalizes well to

---

[†] `https://huggingface.co/mistralai/Mistral-7B-Instruct-v0.3`
[‡] Recommended by ICASSP2023 General Meeting Understanding and Generation Challenge `https://2023.ieeeicassp.org/signal-processing-grand-challenges`.

Table 4: Segmentation performances on three datasets. †: numbers reported in Gao et al. (2023). Other baselines are reported in Jiang et al. (2023d). The best performance is highlighted in **bold**, and the second best is highlighted by underline. Numbers in gray correspond to **supervised** setting.

| Methods | Dialseg711 | | | | SuperDialSeg | | | | TIAGE | | | |
|---|---|---|---|---|---|---|---|---|---|---|---|---|
| | Pk↓ | WD↓ | F1↑ | Score↑ | Pk↓ | WD↓ | F1↑ | Score↑ | Pk↓ | WD↓ | F1↑ | Score↑ |
| **Unsupervised** | | | | | | | | | | | | |
| BayesSeg | 0.306 | 0.350 | 0.556 | 0.614 | 0.433 | 0.593 | 0.438 | 0.463 | 0.486 | 0.571 | 0.366 | 0.419 |
| TextTiling | 0.470 | 0.493 | 0.245 | 0.382 | 0.441 | 0.453 | 0.388 | 0.471 | 0.469 | 0.488 | 0.204 | 0.363 |
| GraphSeg | 0.412 | 0.442 | 0.392 | 0.483 | 0.450 | 0.454 | 0.249 | 0.398 | 0.496 | 0.515 | 0.238 | 0.366 |
| TextTiling+Glove | 0.399 | 0.438 | 0.436 | 0.509 | 0.519 | 0.524 | 0.353 | 0.416 | 0.486 | 0.511 | 0.236 | 0.369 |
| TextTiling+[CLS] | 0.419 | 0.473 | 0.351 | 0.453 | 0.493 | 0.523 | 0.277 | 0.385 | 0.521 | 0.556 | 0.218 | 0.340 |
| TextTiling+NSP | 0.347 | 0.360 | 0.347 | 0.497 | 0.512 | 0.521 | 0.208 | 0.346 | 0.425 | 0.439 | 0.285 | 0.426 |
| GreedySeg | 0.381 | 0.410 | 0.445 | 0.525 | 0.490 | 0.494 | 0.365 | 0.437 | 0.490 | 0.506 | 0.181 | 0.341 |
| CSM | 0.278 | 0.302 | 0.610 | 0.660 | 0.462 | 0.467 | 0.381 | 0.458 | 0.400 | 0.420 | 0.427 | 0.509 |
| DialSTART † | 0.178 | 0.198 | - | - | - | - | - | - | - | - | - | - |
| **Ours (zero-shot)** | **0.093** | **0.103** | **0.888** | **0.895** | **0.277** | **0.289** | **0.758** | **0.738** | **0.363** | **0.401** | **0.596** | **0.607** |
| **Transfer from TIAGE to Target** | | | | | | | | | | | | |
| TextSeg$_{dial}$ | 0.476 | 0.491 | 0.182 | 0.349 | 0.552 | 0.570 | 0.199 | 0.319 | 0.357 | 0.386 | 0.450 | 0.539 |
| BERT | 0.441 | 0.411 | 0.005 | 0.297 | 0.511 | 0.513 | 0.043 | 0.266 | 0.418 | 0.435 | 0.124 | 0.349 |
| RoBERTa | 0.197 | 0.210 | 0.650 | 0.723 | 0.434 | 0.436 | 0.276 | 0.420 | **0.265** | **0.287** | 0.572 | **0.648** |
| **Ours (w/ reflection)** | **0.050** | **0.056** | **0.921** | **0.934** | **0.265** | **0.273** | **0.765** | **0.748** | 0.333 | 0.362 | **0.632** | 0.642 |
| **Transfer from SuperDialseg to Target** | | | | | | | | | | | | |
| TextSeg$_{dial}$ | 0.453 | 0.461 | 0.367 | 0.455 | 0.199 | 0.204 | 0.760 | 0.779 | 0.489 | 0.508 | 0.266 | 0.384 |
| BERT | 0.401 | 0.473 | 0.381 | 0.472 | 0.214 | 0.225 | 0.725 | 0.753 | 0.492 | 0.526 | 0.226 | 0.359 |
| RoBERTa | 0.241 | 0.272 | 0.660 | 0.702 | **0.185** | **0.192** | **0.784** | **0.798** | 0.401 | 0.418 | 0.373 | 0.482 |
| **Ours (w/ reflection)** | **0.049** | **0.054** | **0.924** | **0.936** | 0.256 | 0.264 | 0.776 | 0.758 | **0.318** | **0.345** | **0.634** | **0.651** |

the target dataset, surpassing the baseline model trained on the full source training set and even outperforming some supervised baselines.

# 4 RELATED WORKS

## 4.1 MEMORY MANAGEMENT IN CONVERSATION

Long-term open-domain conversation (Feng et al., 2020; Xu et al., 2022; Maharana et al., 2024) poses significant challenges for LLM-powered conversational agents. To address this, memory management (Lu et al., 2023; Wang et al., 2023; Zhong et al., 2024; Wu et al., 2024; Li et al., 2024; Zhang et al., 2024) is widely adopted. The core of memory management involves leveraging dialogue history to provide background information, extract persona, understand the user's intent, and generate history-aware responses. For instance, MPC (Lee et al., 2023), MemoryBank (Zhong et al., 2024) and COMEDY (Chen et al., 2024) further summarize past events in the conversation history as memory records. Methods such as RecurSum (Wang et al., 2023) and ConditionMem (Yuan et al., 2023) consider the memory updating process through recursive summarization.

Inspired by the success of retrieval-augmented generation (RAG), many recent works introduce retrieval modules into memory management. For example, MSC (Xu et al., 2022) utilizes a pre-trained Dense Passage Retriever (DPR) (Karpukhin et al., 2020) model to select the top $N$ relevant summaries. Instead of using a retrieval model, MemoChat (Lu et al., 2023) employs an LLM to retrieve relevant memory records. Recently, Maharana et al. (2024) release a dataset, *LOCOMO*, which is specifically designed to assess long-term conversational memory, highlighting the effectiveness of RAG in maintaining long-term memory. Their experiment results indicate that long-context LLMs are prone to generating hallucinations, and summary-only memory results in sub-optimal performance due to information loss.

## 4.2 Chunking Granularity in RAG System

Chunking granularity (Duarte et al., 2024) (i.e., how the entire context is segmented into retrieval units) is a crucial aspect of RAG systems. Ineffective segmentation can result in incomplete or noisy retrieval units, which can impair the retrieval module (Yu et al., 2023) and negatively impact the subsequent response generation (Shi et al., 2023).

Semantic-based chunking strategies (Mishra, 2023; Antematter Team, 2024; Greg Kamradt, 2024) use representation similarity to identify topic shifts and decide chunk boundaries. With the advancement of LLMs, some studies leverage their capabilities to segment context into retrieval units. For instance, LumberChunker (Duarte et al., 2024) segments narrative documents into semantically coherent chunks using Gemini (Team et al., 2023). However, existing research mainly focuses on document chunking, overlooking conversation chunking. Common chunking practices (LangChain Team, 2023b; LlamaIndex Team, 2023) in conversations directly rely on the natural structure (*i.e.,* utterances or dialogue turns) of dialogue to divide conversation into retrieval units.

## 4.3 Denoising in RAG system

Recent studies have observed that noise in conversations can negatively impact the retrieval module in RAG systems. For example, COTED (Mao et al., 2022) found that redundant noise in dialogue rounds significantly impairs conversational search. Earlier research (Strzalkowski et al., 1998; Wasson, 2002) investigates the use of summaries in retrieval systems. With the advent of LLM, recent approaches (Ravfogel et al., 2023; Lee et al., 2024) denoise raw dialogues by prompting LLMs to summarize. Subsequently, they fine-tune the retriever's embedding model to align vector representations of original text with those of generated summaries. However, these methods have several drawbacks: (1) summarization introduces latency and computational costs, whereas dialogue state methods require high-quality annotated data. (2) Fine-tuning the retriever's embedding model limits flexibility and scalability, restricting it from being used as a plug-and-play method. (3) Fine-tuning risks overfitting and catastrophic forgetting (McCloskey & Cohen, 1989; Lee et al., 2022), potentially impeding domain adaptation and generalization ability of pre-trained retrievers.

## 5 Conclusion

In this paper, we systematically investigate the impact of memory granularity on retrieval-augmented response generation for long-term conversational agents. Our findings reveal the limitations of turn-level and session-level memory granularities, as well as summarization-based methods. To overcome these challenges, we introduce SeCom, a novel memory management system that constructs a memory bank at the segment-level and employs compression-based denoising techniques to enhance retrieval performance. The experimental results underscore the effectiveness of SeCom in handling long-term conversations. Further analysis and ablation studies confirm the contributions of the segment-level memory units and the compression-based denoising technique within our framework.

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

# A    APPENDIX

## A.1    DETAILS OF CONVERSATION SEGMENTATION MODEL

We use GPT-4-0125 as the backbone LLM for segmentation. The zero-shot segmentation prompt is provided in Figure 6. It instructs the segmentation model to generate all segmentation indices at once, avoiding the iterative segmentation process used in LumberChunker (Duarte et al., 2024), which can lead to unacceptable latency. We specify that the output should be in **JSONL** format to facilitate subsequent processing. To generate segmentation guidance, we select the top 100 poorly segmented samples with the largest Window Diff metric from the training set. The segmentation guidance consists of two parts: (1) *Segmentation Rubric*: Criteria items on how to make better segmentation. (2) *Representative Examples*: The most representative examples that include the ground-truth segmentation, the model's prediction, and the reflection on the model's errors. The number of rubric items is set to 10. To meet this requirement, we divide the top 100 poorly segmented samples into 10 mini-batches and prompt the LLM-based segmentation model to reflect on each batch individually. The segmentation model is also asked to select the most representative example in each batch, which is done concurrently with rubric generation. Figure 7 presents the prompt used to generate rubric. The generated rubric is shown at Fig. 9 and Fig. 10 on *TIAGE* and *SuperDialSeg*, respectively. After the segmentation guidance is learned, we utilize the prompt shown in Figure 8 as a few-shot segmentation prompt. For simplicity and fair comparison, we do not use any rubric for conversation segmentation in *LOCOMO* and *Long-MT-Bench+*.

## A.2    ADDITIONAL COST ANALYSIS

Table 5 compares the overall costs involved in memory construction, memory retrieval, and response generation across different methods. The results demonstrate that our method significantly enhances performance compared to the baseline while only slightly increasing computational overhead, and it outperforms the MemoChat method in both efficiency and effectiveness.

## A.3    THE ANALOGY BETWEEN THE REFLECTION AUGMENTATION AND PREFIX-TUNING

When a small amount of conversation data with segment annotations is available, we explore how to leverage this data to transfer segmentation knowledge and better align the LLM-based segmentation model with human preferences. Inspired by the prefix-tuning technique (Li & Liang, 2021) and reflection mechanism (Shinn et al., 2023; Renze & Guven, 2024), we treat the segmentation prompt as the "prefix" and iteratively optimize it through LLM self-reflection, ultimately obtaining a segmentation guidance $G$.

Table 5: Comparison between our method and *MemoChat* from multiple aspects on *Long-MT-Bench+*. "# In. Token", "# Out. Token" and "Latency" report the number of input / output token and the latency per question, including memory construction, memory retrieval and reponse generation.

| Methods | # In. Token | # Out. Token | Latency (s) | GPT Score |
|---|---|---|---|---|
| Session-Level | 3,642 | 102 | 2.17 | 73.38 |
| MemoChat | 7,233 | 229 | 5.60 | 85.14 |
| Ours | 1,722 | 135 | 2.61 | **88.81** |

---

**Instruction Part of the Segmentation Prompt (Zero-Shot).**

```
# Instruction
## Context
- **Goal**: Your task is to segment a multi-turn conversation between a
user and a chatbot into topically coherent units based on semantics.
Successive user-bot exchanges with the same topic should be grouped
into the same segmentation unit, and new segmentation units should
be created when topic shifts.
- **Data**: The input data is a series of user-bot exchanges separated
by "\n\n". Each exchange consists of a single-turn conversation between
the user and the chatbot, started with "[Exchange (Exchange Number)]: ".
### Output Format
- Output the segmentation results in **JSONL (JSON Lines)** format.
Each dictionary represents a segment, consisting of one or more
user-bot exchanges on the same topic.
Each dictionary should include the following keys:
   - **segment_id**: The index of this segment, starting from 0.
   - **start_exchange_number**: The number of the **first** user-bot
   exchange in this segment.
   - **end_exchange_number**: The number of the **last**
   user-bot exchange in this segment.
   - **num_exchanges**: An integer indicating the number of
   user-bot exchanges in this segment, calculated as:
   **end_exchange_number** - **start_exchange_number** + 1.
Here is an example of the expected output:
```
<segmentation>
{"segment_id": 0, "start_exchange_number": 0,
"end_exchange_number": 5, "num_exchanges": 6}
{"segment_id": 1, "start_exchange_number": 6,
"end_exchange_number": 8, "num_exchanges": 3}
...
</segmentation>
```
# Data
{{text_to_be_segmented}}
# Question
## Please generate the segmentation result from the input data that
meets the following requirements:
- **No Missing Exchanges**:  Ensure that the exchange numbers cover
all exchanges in the given conversation without omission.
- **No Overlapping Exchanges**: Ensure that successive segments have
no overlap in exchanges.
- **Accurate Counting**:  The sum of **num_exchanges**
across all segments should equal the total number of user-bot exchanges.
- Provide your segmentation result between the tags:
<segmentation></segmentation>.
# Output
Now, provide the segmentation result based on the instructions above.
```

Figure 6: Prompt for GPT-4 segmentation (zero-shot).

Prefix-tuning seeks to learn a prefix matrix $\boldsymbol{P}$ to boost the performance of the language model $\mathrm{LM}_\phi$ without fine-tuning its parameter $\phi$. The prefix matrix $\boldsymbol{P}$ is prepended to the activation $h$ of the Transformer layer:

$$h_i = \begin{cases} \boldsymbol{P}[i,:], & \text{if } i \in \mathcal{P}_{idx} \\ \mathrm{LM}_\phi\left(z_i, h_{<i}\right), & \text{otherwise} \end{cases} \tag{4}$$

where $\mathcal{P}_{idx}$ is the prefix indices.

---

**Prompt for Generating the Segmentation Guidance**

```
# Instruction
## Context
**Goal**: Your task is to evaluate the differences between a language
model's predicted segmentation and the ground-truth segmentation made
by expert annotators for multiple human-bot conversations.
Analyze these differences, reflect on the prediction errors, and
generate one concise rubric item for future conversation segmentation.
You will be provided with some existing rubric items derived
from previous examples.
1. Begin by reviewing and copying the existing rubric items.
2. Modify, update, or replace the existing items if they do not
adequately address the current segmentation errors.
3. Generate only one new rubric item to minimize segmentation errors
in the given examples.
4. Select and reflect on the most representative example
from the provided data.
**Data**: You will receive a segmented conversation example,
including both the prediction and the ground-truth segmentation.
Each segment begins with "Segment segment_id:".
Additionally, you will be provided with some existing rubric items
derived from previous examples. Modify, update, or even replace them
if they do not adequately explain the current segmentation mistakes.
## Requirements
- Add at most one new rubric item at a time even
though multiple examples are provided.
- Ensure the rubric is user-centric, concise, and each item
is mutually exclusive.
- You can modify, update, or replace the existing items
if they do not adequately
address the current segmentation errors.
- Present your new rubric item within `<rubric></rubric>`.
- Provide the most representative example with your reflection
within `<example></example>`. Here is an example:
```
<reflection>
Your reflection on the prediction errors,
example by example.
</reflection>
<rubric>
- [one and only one new rubric item]
</rubric>
<example>
Present the most representative example,
along with your reflection on this example.
</example>
```
# Existing Rubric: {{past_rubric}}
# Examples: {{examples}}

# Output
```

Figure 7: Prompt for generating segmentation guidance.

In the context of our segmentation scenario, our goal is to "learn" a textual guidance $G$ that directs the segmentation model toward improved segmentation outcomes. The process of updating the segmentation guidance $G$ parallels the optimization of the prefix parameter $P$ in prefix-tuning. Initially, the segmentation guidance $G_0$ is set to empty, analogous to the initial prefix parameter $P_0$. During each iteration of guidance updating, we first apply our conversation segmentation model in a zero-shot manner to a batch of conversation data. Building upon the insights that LLMs possess the ability for self-reflection and improvement (Shinn et al., 2023; Renze & Guven, 2024), we then

Table 6: Performance comparison on the official question-answer pairs of *LOCOMO* using MPNet retriever. All other settings remain the same as in Table 1. MemoChat (Lu et al., 2023) is not applicable in *Mistral-7B-Instruct-v0.3* due to Mistral's inability to execute the "Memo Writing" step, as it often fails to generate a valid JSON response needed to construct the memory bank in Lu et al. (2023).

| Methods | QA Performance | | | | | | Context Length | |
|---|---|---|---|---|---|---|---|---|
| | GPT4Score | BLEU | Rouge1 | Rouge2 | RougeL | BERTScore | # Turns | # Tokens |
| *GPT-35-Turbo* | | | | | | | | |
| Full History | 66.28 | 7.51 | 28.73 | 14.07 | 27.90 | 87.82 | 293 | 18,655 |
| MemoChat | 75.77 | 11.28 | 32.91 | 18.82 | 29.78 | 87.98 | - | 1,159 |
| Turn-Level | 81.52 | 11.91 | 36.00 | 19.59 | 34.99 | **88.64** | 55.00 | 3,026 |
| Session-Level | 74.20 | 10.95 | 29.92 | 14.64 | 29.27 | 87.96 | 54.48 | 3,442 |
| SECOM | **84.21** | **12.80** | **36.70** | **19.90** | **35.61** | 88.59 | 56.49 | 3,565 |
| *Mistral-7B-v0.3* | | | | | | | | |
| Full History | 69.13 | 6.77 | 30.40 | 15.02 | 29.20 | 87.29 | 293 | 18,655 |
| Turn-Level | 78.82 | 10.09 | 32.75 | 16.25 | 31.75 | **87.97** | 55.00 | 3,026 |
| Session-Level | 62.68 | 7.37 | 26.68 | 12.38 | 25.86 | 86.98 | 54.48 | 3,442 |
| SECOM | **80.07** | **10.67** | **32.82** | **16.65** | **31.81** | 87.87 | 56.49 | 3,565 |

instruct the segmentation model to reflect on its mistakes given the ground-truth segmentation and update the segmentation guidance $G$. This process mirrors Stochastic Gradient Descent (SGD) optimization:

$$G_{m+1} = G_m - \eta \nabla \mathcal{L} (G_m),\tag{5}$$

where $\nabla \mathcal{L} (G_m)$ denotes the gradient of segmentation loss, which we assume is estimated implicitly by the LLM itself and used to adjust the next segmentation guidance $G_{m+1}$.

### A.4 PROMPT FOR GPT-4 EVALUATION

We use the same evaluation prompts as MemoChat (Lu et al., 2023). The LLM-powered evaluation consists of single-sample scoring (GPT4Score) and pair-wise comparison. The evaluation prompts are displayed in Figure 12. For pair-wise comparison, we alternate the order of the responses and conduct a second comparison for each pair to minimize position bias.

### A.5 EVALUATION RESULTS ON THE OFFICIAL QA PAIRS OF LOCOMO

As *LOCOMO* (Maharana et al., 2024) released a subset containing QA pairs recently. To ensure reproducibility, we evaluate our method on these official QA pairs. Table 6 presents the evaluation results. The superiority of our SECOM is also evident on these QA pairs, demonstrating its superior effectiveness and robustness.

### A.6 CASE STUDY

To further demonstrate the advantages of our method, we conduct a qualitative evaluation. Figure 13 presents a specific case comparing the segment-level memory with the turn-level memory. It demonstrates that using turn-level memory units fails to address the user's request. We attribute this to the fragmentation of user-agent turns, and the critical turns may not explicitly contain or relate to the keywords in the user's request.

Similarly, using session-level memory units is also sub-optimal, as illustrated in Figure 14. This issue arises because a session often includes multiple topics, introducing a significant amount of irrelevant information that hampers effective retrieval. The irrelevant information also distracts the LLM, as noted in previous studies (Shi et al., 2023; Liu et al., 2024).

We also conduct a case study to compare our method with two recent, powerful memory management techniques: *RecurSum* (Wang et al., 2023) and *ConditionMem* (Yuan et al., 2023), as shown in Figure 15 and Figure 16. The results indicate that the summarization process in these methods often omits detailed information that is essential for accurately answering the user's request.

Table 7: Statistics of the *MT-Bench+* and the constructed *Long-MT-Bench+* datasets. The notation *"# Item"* represents the average number of the corresponding item per conversation.

| Datasets | # QA. Pairs | # Session | # Round | # Token |
|---|---|---|---|---|
| MT-Bench+ | 1 | 1 | 13.33 | 3,929 |
| Long-MT-Bench+ | 26.09 | 4.91 | 65.45 | 19,287 |

### A.7 DETAILS OF DATASET CONSTRUCTION

(i) *LOCOMO* (Maharana et al., 2024): this dataset contains the longest conversations to date, with an average of more than 9K tokens per sample. Since *LOCOMO* does not release the corresponding question-answer pairs when we conduct our experiment, we prompt GPT-4 to generate QA pairs for each session as in Alonso et al. (2024). We also conduct evaluation on the recently released official QA pairs in Appendix A.5.

(ii) *Long-MT-Bench+*: *Long-MT-Bench+* is reconstructed from the *MT-Bench+* (Lu et al., 2023) dataset. In *MT-Bench+*, human experts are invited to expand the original questions and create long-range questions as test samples. However, there are two drawbacks when using this dataset to evaluate the memory mechanism of conversational agents: (1) the number of QA pairs is relatively small, with only 54 human-written long-range questions; and (2) the conversation length is not sufficiently long, with each conversation containing an average of 13.3 dialogue turns and a maximum of 16 turns. In contrast, the conversation in *LOCOMO* has an average of 300 turns and 9K tokens. To address (1), we use these human-written questions as few-shot examples and ask GPT-4 to generate a long-range test question for each dialogue topic. For (2), following (Yuan et al., 2023), we merge five consecutive sessions into one, forming longer dialogues that are more suitable for evaluating memory in long-term conversation. We refer to the reconstructed dataset as *Long-MT-Bench+* and present its statistics in Table 7.

### A.8 DETAILS OF RETRIEVAL PERFORMANCE MEASUREMENT

We measure the retrieval performance in terms of the discounted cumulative gain (DCG) metric (Järvelin & Kekäläinen, 2002):

$$DCG = \sum_{i=1}^{p} \frac{rel_i}{\log_2(i+1)}, \tag{6}$$

where $rel_i$ denotes the relevance score of the retrieved user-agent turn ranked at position $i$, and $p$ represents the total number of retrieved turns. Note that in the *Long-MT-Bench+* dataset, answering a single question often requires referring to several consecutive turns. Therefore, we distribute the relevance score evenly across these relevant turns and set the relevance score of irrelevant turns to zero. For instance, assume that the ground truth reference turn set for question $q$ is $\mathcal{R}(q) = \{r_{k+j}\}_{j=1}^{N}$, which is provided by the dataset. In this case, the relevance score for each turn is set as follows:

$$rel_i = \begin{cases} 0 & i < k+1 \\ \frac{1}{N} & k+1 \le i \le k+N \\ 0 & i > k+N \end{cases}.$$

This approach allows us to evaluate retrieval performance at different granularity.

### A.9 ADDITIONAL EXPERIMENTS ON COQA AND PERSONA-CHAT

To further validate SeCom's robustness and versatility across a broader range of dialogue types, we conduct additional experiments on other benchmarks, **Persona-Chat** (Zhang et al., 2018) and **CoQA** (Reddy et al., 2019).

Given the relatively short context length of individual samples in these datasets, we adopt an approach similar to Long-MT-Bench+ by aggregating multiple adjacent samples into a single instance. For CoQA, each sample is supplemented with the text passages of its 10 surrounding samples. Since CoQA answers are derived from text passages rather than dialogue turns, we replace the turn-level

Table 8: QA performance comparison on *CoQA* using MPNet-based retrieval model. The response generation model is `GPT-3.5-Turbo`.

| Methods | GPT4Score | BLEU | Rouge1 | Rouge2 | RougeL | BERTScore | #Tokens |
|---|---|---|---|---|---|---|---|
| Sentence-Level | 95.55 | 36.02 | 48.58 | 37.96 | 47.03 | 90.01 | 993 |
| Session-Level | 91.58 | 31.22 | 47.18 | 37.32 | 45.92 | 89.65 | 3,305 |
| ConditionMem | 94.32 | 34.35 | 47.91 | 37.55 | 46.38 | 89.77 | 1,352 |
| MemoChat | 97.16 | 38.17 | 49.54 | 38.23 | 47.77 | 90.14 | 1,041 |
| COMEDY | 97.48 | 38.02 | 49.41 | 38.19 | 47.63 | 90.06 | 3,783 |
| SECOM (Ours) | **98.31** | **39.57** | **50.44** | **39.51** | **48.98** | **90.37** | 1,016 |

Table 9: Next utterance prediction performance comparison on *Persona-Chat* using MPNet-based retrieval model. The response generation model is `GPT-3.5-Turbo`.

| Methods | Performance | | | | | | Context Length | |
|---|---|---|---|---|---|---|---|---|
| | GPT4Score | BLEU | Rouge1 | Rouge2 | RougeL | BERTScore | # Turns | # Tokens |
| Turn-Level | 69.23 | 5.73 | 21.38 | 9.06 | 19.87 | 87.28 | 24.00 | 682 |
| Session-Level | 67.35 | 5.45 | 21.80 | 8.86 | 20.04 | 87.34 | 116.91 | 3,593 |
| ConditionMem | 73.21 | 6.16 | 22.52 | 9.88 | 20.95 | 87.44 | - | 1,388 |
| MemoChat | 76.83 | 7.21 | 25.13 | 10.81 | 22.31 | 87.68 | - | 1,296 |
| COMEDY | 76.52 | 7.05 | 24.97 | 10.54 | 22.18 | 87.60 | - | 3,931 |
| SECOM (Ours) | **78.34** | **7.75** | **26.01** | **11.57** | **23.98** | **87.82** | 23.48 | 702 |

baseline with a sentence-level baseline. For Persona-Chat, we utilize the expanded version provided by Jandaghi et al. (2023). Conversations are aggregated by combining each sample with its 5 surrounding samples. Following the next utterance prediction protocol, we include the personas of both conversational roles in the prompt. Due to the large scale of these datasets, we select subsets for experimentation. From CoQA, we randomly sample 50 instances from an initial pool of 500, resulting in a subset containing over 700 QA pairs. Similarly, for Persona-Chat, we randomly select 100 instances, encompassing over 1,000 utterances in total.

As shown in Table 8 and Table 9, SECOM consistently outperforms baseline methods across these datasets, highlighting its effectiveness in handling diverse dialogue scenarios, including open-ended and multi-turn interactions.

## A.10 HUMAN EVALUATION RESULTS

To ensure a holistic assessment, we conduct human evaluation to gauge the quality of the LLM's response in conversation. We adopt the human evaluation scheme of COMEDY (Chen et al., 2024), which encompasses five perspectives: Coherence, Consistency, Engagingness, Humanness and Memorability. Ten human annotators are asked to score the responses following a detailed rubric for each perspective. Results in Table 10 show that the rank of different methods from human evaluation is generally consistent with those obtained from automated metrics, confirming the practical effectiveness of our proposed approach.

## A.11 PERFORMANCE USING SMALLER SEGMENTATION MODEL

To make our method applicable in resource-constrained environments, we conduct additional experiments by replacing the `GPT-4-Turbo` used for the segmentation model with the `Mistral-7B-Instruct-v0.3` and a `RoBERTa` based model fine-tuned on SuperDialseg (Jiang et al., 2023d). Table 11 shows that SECOM maintains the advantage over baseline methods when switching from GPT-4 to Mistral-7B. Notably, even with a RoBERTa based segmentation model, SECOM retains a substantial performance gap over other granularity-based baselines.

Table 10: Human evaluation results on *Long-MT-Bench+* using MPNet-based retrieval model. The response generation model is `GPT-3.5-Turbo`.

| Methods | Coherence | Consistency | Memorability | Engagingness | Humanness | Average |
|---|---|---|---|---|---|---|
| Full-History | 1.55 | 1.11 | 0.43 | 0.33 | 1.85 | 1.05 |
| Sentence-Level | 1.89 | 1.20 | 1.06 | 0.78 | 2.00 | 1.39 |
| Session-Level | 1.75 | 1.25 | 0.98 | 0.80 | 1.92 | 1.34 |
| ConditionMem | 1.58 | 1.08 | 0.57 | 0.49 | 1.77 | 1.10 |
| MemoChat | 2.05 | 1.25 | 1.12 | 0.86 | **2.10** | 1.48 |
| COMEDY | **2.20** | 1.28 | 1.20 | 0.90 | 1.97 | 1.51 |
| SECOM (Ours) | 2.13 | **1.34** | **1.28** | **0.94** | 2.06 | 1.55 |

Table 11: Performance comparison on *LOCOMO* and *Long-MT-Bench+* using different segmentation model. The retriever is MPNet-based and other settings follow Table 1.

| Methods | QA Performance | | | | | | Context Length | |
|---|---|---|---|---|---|---|---|---|
| | GPT4Score | BLEU | Rouge1 | Rouge2 | RougeL | BERTScore | # Turns | # Tokens |
| *LOCOMO* | | | | | | | | |
| Zero History | 24.86 | 1.94 | 17.36 | 3.72 | 13.24 | 85.83 | 0.00 | 0 |
| Full History | 54.15 | 6.26 | 27.20 | 12.07 | 22.39 | 88.06 | 210.34 | 13,330 |
| Turn-Level (MPNet) | 57.99 | 6.07 | 26.61 | 11.38 | 21.60 | 88.01 | 54.77 | 3,288 |
| Session-Level (MPNet) | 51.18 | 5.22 | 24.23 | 9.33 | 19.51 | 87.45 | 53.88 | 3,471 |
| SumMem | 53.87 | 2.87 | 20.71 | 6.66 | 16.25 | 86.88 | - | 4,108 |
| RecurSum | 56.25 | 2.22 | 20.04 | 8.36 | 16.25 | 86.47 | - | 400 |
| ConditionMem | 65.92 | 3.41 | 22.28 | 7.86 | 17.54 | 87.23 | - | 3,563 |
| MemoChat | 65.10 | 6.76 | 28.54 | 12.93 | 23.65 | 88.13 | - | 1,159 |
| **SECOM** (RoBERTa-Seg) | 61.84 | 6.41 | 27.51 | 12.27 | 23.06 | 88.08 | 56.32 | 3,767 |
| **SECOM** (Mistral-7B-Seg) | 66.37 | 6.95 | 28.86 | 13.21 | 23.96 | 88.27 | 55.80 | 3,720 |
| **SECOM** (GPT-4-Seg) | **69.33** | **7.19** | **29.58** | **13.74** | **24.38** | **88.60** | 55.51 | 3,716 |
| *Long-MT-Bench+* | | | | | | | | |
| Zero History | 49.73 | 4.38 | 18.69 | 6.98 | 13.94 | 84.22 | 0.00 | 0 |
| Full History | 63.85 | 7.51 | 26.54 | 12.87 | 20.76 | 85.90 | 65.45 | 19,287 |
| Turn-Level (MPNet) | 84.91 | 12.09 | 34.31 | 19.08 | **27.82** | 86.49 | 3.00 | 909 |
| Session-Level (MPNet) | 73.38 | 8.89 | 29.34 | 14.30 | 22.79 | 86.61 | 13.43 | 3,680 |
| SumMem | 63.42 | 7.84 | 25.48 | 10.61 | 18.66 | 85.70 | - | 1,651 |
| RecurSum | 62.96 | 7.17 | 22.53 | 9.42 | 16.97 | 84.90 | - | 567 |
| ConditionMem | 63.55 | 7.82 | 26.18 | 11.40 | 19.56 | 86.10 | - | 1,085 |
| MemoChat | 85.14 | 12.66 | 33.84 | 19.01 | 26.87 | 87.21 | - | 1,615 |
| **SECOM** (RoBERTa-Seg) | 81.52 | 11.27 | 32.66 | 16.23 | 25.51 | 86.63 | 2.96 | 841 |
| **SECOM** (Mistral-7B-Seg) | 86.32 | 12.41 | 34.37 | 19.01 | 26.94 | 87.43 | 2.85 | 834 |
| **SECOM** (GPT-4-Seg) | **88.81** | **13.80** | **34.63** | **19.21** | 27.64 | **87.72** | 2.77 | 820 |

Instruction Part of the Segmentation Prompt (W/ Reflection).

```
# Instruction
## Context
- **Goal**: Your task is to segment a multi-turn conversation between a
user and a chatbot into topically coherent units based on semantics.
Successive user-bot exchanges with the same topic should be grouped
into the same segmentation unit, and new segmentation units should
be created when topic shifts.
- **Data**: The input data is a series of user-bot exchanges separated
by "\n\n". Each exchange consists of a single-turn conversation between
the user and the chatbot, started with "[Exchange (Exchange Number)]: ".
- **Tips**: Refer fully to the provided rubric
and examples for guidance on segmentation.
## Requirements
### Output Format
- Output the segmentation results in **JSONL (JSON Lines)** format.
Each dictionary represents a segment, consisting of one or more
user-bot exchanges on the same topic.
Each dictionary should include the following keys:
  - **segment_id**: The index of this segment, starting from 0.
  - **start_exchange_number**: The number of the **first** user-bot
  exchange in this segment.
  - **end_exchange_number**: The number of the **last**
  user-bot exchange in this segment.
  - **num_exchanges**: An integer indicating the number of
  user-bot exchanges in this segment, calculated as:
  **end_exchange_number** - **start_exchange_number** + 1.
Here is an example of the expected output:
```
<segmentation>
{"segment_id": 0, "start_exchange_number": 0,
"end_exchange_number": 5, "num_exchanges": 6}
{"segment_id": 1, "start_exchange_number": 6,
"end_exchange_number": 8, "num_exchanges": 3}
...
</segmentation>
```
## Segment Rubric
{{segment_rubric}}
## Segment Examples
{{segment_examples}}
# Data
{{text_to_be_segmented}}
# Question
## Please generate the segmentation result from the input data that
meets the following requirements:
- **No Missing Exchanges**:  Ensure that the exchange numbers cover
all exchanges in the given conversation without omission.
- **No Overlapping Exchanges**: Ensure that successive segments have
no overlap in exchanges.
- **Accurate Counting**:  The sum of **num_exchanges**
across all segments should equal the total number of user-bot exchanges.
- **Utilize Segment Rubric**: Use the given segment rubric
and examples to better segment.
- Provide your segmentation result between the tags:
<segmentation></segmentation>.
# Output
Now, provide the segmentation result based on the instructions above.
```

Figure 8: Prompt for GPT-4 segmentation (w/ reflection).

**Segmentation rubric learned from *TIAGE***

- Ensure segments encapsulate a complete thematic or topical exchange before initiating a new segment. This includes recognizing when a topic shift is part of the same thematic exchange and should not trigger a new segment.

- Segments should not only capture the flow of conversation by recognizing subtle topic shifts but also ensure that related questions and answers, or setup and response exchanges, are included within the same segment to preserve the natural flow and context of the dialogue.

- Maintain the integrity of conversational dynamics, ensuring that exchanges which include setup and response (or question and answer) are not divided across segments. This preserves the context and flow of the dialogue, recognizing that some topic shifts, while apparent, are part of a larger thematic discussion.

- Segments must accurately reflect the thematic depth of the conversation, ensuring that all parts of a thematic exchange, including indirect responses or tangentially related comments, are grouped within the same segment to maintain conversational coherence.

- Evaluate the conversational cues and context to determine the thematic linkage between exchanges. Avoid creating new segments for responses that, while seemingly off-topic, are contextually related to the preceding messages, ensuring a coherent and unified thematic narrative.

- Prioritize the preservation of conversational momentum when determining segment boundaries, ensuring that the segmentation does not interrupt the natural progression of dialogue or the development of thematic elements, even when the conversation takes unexpected turns.

- Assess the thematic relevance of each conversational turn, ensuring segments are not prematurely divided by superficial topic changes that are part of a broader thematic dialogue. This includes recognizing when a seemingly new topic is a direct continuation or an elaboration of the previous exchange, thereby maintaining thematic coherence and conversational flow.

- Consider the conversational and thematic continuity over superficial changes in topic or structure when segmenting conversations. This ensures that segments reflect the natural flow and thematic integrity of the dialogue, even when the conversation takes subtle turns.

- Incorporate flexibility in segment boundaries to accommodate for the natural ebb and flow of conversational topics, ensuring that segments are not overly fragmented by minor topic shifts that remain within the scope of the overarching thematic dialogue.

- Avoid over-segmentation by recognizing the thematic bridges between conversational turns. Even when a conversation appears to shift topics, if the underlying theme or narrative purpose connects the exchanges, they should be considered part of the same segment to preserve the dialogue's natural progression and thematic integrity.

Figure 9: Segmentation rubric learned on *TIAGE* (Xie et al., 2021).

**Segmentation rubric learned from *SuperDialSeg***

- Segmentation should reflect natural pauses or shifts in the conversation, indicating a change in topic or focus.
- Each segment should aim to be self-contained, providing enough context for the reader to understand the topic or question being addressed without needing to refer to other segments.
- Ensure segmentation captures the full scope of a thematic exchange, using linguistic cues and conversational context to guide the identification of natural breaks or transitions in dialogue.
- Segmentation should prioritize thematic continuity over structural cues alone, ensuring that all parts of a thematic exchange, including follow-up questions or clarifications, are contained within the same segment.
- Segments must ensure logical and thematic coherence, grouping together all elements of an exchange that contribute to a single topic or question, even if the conversation appears structurally disjointed.
- Ensure segments maintain thematic progression, especially in conversations where multiple inquiries and responses explore different facets of the same overarching topic.
- Segmentation should avoid over-segmentation by ensuring that a series of inquiries and responses that explore different aspects of a single overarching topic are grouped within the same segment, even if they contain multiple question-answer pairs.
- Ensure that segments are not prematurely divided based on superficial structural cues like greetings or sign-offs, but rather on the substantive thematic content of the exchange.
- Ensure segmentation recognizes and preserves the thematic progression within a conversation, even when minor topic shifts occur, by evaluating the overall context and goal of the exchange rather than segmenting based on immediate linguistic cues alone.
- Ensure that segments accurately reflect the inquiry-response cycle, grouping all related questions and their corresponding answers into a single segment to preserve the flow and coherence of the conversation.

Figure 10: Segmentation rubric learned on *SuperDialSeg* (Jiang et al., 2023d).

**Ground-truth Segment:**

- **Segment 0:** hello, how are you doing? hello. pretty good, thanks. and yourself? awesome, i just got back from a bike ride. cool! do you spend a lot of time biking? yup. its my favorite thing to do. do you? i love playing folk music. i actually hope to be a professional musician someday that is interesting. what instruments do you play? i can play the guitar and the piano and i also like to sing. i can only sing when i drink, but i do not like to do that anymore.

- **Segment 1:** i m not a big drinker either. do you have a job? construction, like my dad. what do you do when you are not being a rock star nice! i work as a custodian. not too glamorous but it pays the bills haha i feel ya. you gotta do what you gotta do. exactly. do you have other hobbies besides biking?

**Predicted Segment:**

- **Segment 0:** hello, how are you doing? hello. pretty good, thanks. and yourself? awesome, i just got back from a bike ride.

- **Segment 1:** cool! do you spend a lot of time biking? yup. its my favorite thing to do. do you? i love playing folk music. i actually hope to be a professional musician someday

- **Segment 2:** that is interesting . what instruments do you play? i can play the guitar and the piano and i also like to sing. i can only sing when i drink, but i do not like to do that anymore.

- **Segment 3:** i m not a big drinker either. do you have a job? construction, like my dad. what do you do when you are not being a rock star nice! i work as a custodian. not too glamorous but it pays the bills haha

- **Segment 4:** i feel ya. you gotta do what you gotta do. exactly. do you have other hobbies besides biking?

Figure 11: An example of poor segmentation from GPT-4 zero-shot segmentation illustrates that the GPT-4 powered segmentation model favors a more fine-grained segmentation. The Window Diff metric between the ground truth and the prediction is 0.80.

**Single-Sample Score**

```
You are an impartial judge. You will be shown Related
Conversation History, User Question and Bot Response.
'''\nRelated Conversation History\nRCH\_0\n'''
'''\nUser Question\nUQ\_1\n'''
'''\nBot Response\nBR\_2\n'''
Please evaluate whether Bot Response is faithful to the content of
Related Conversation History to answer User Question.
Begin your evaluation by providing a short explanation,
then you must rate Bot Response on an integer rating of 1 to
100
by strictly following this format:
<rating>an integer rating of 1 to 100</rating>.
```

**Pair-Wise Comparison**

```
You are an impartial judge. You will be shown
Related Conversation History, User Question and Bot Response.
'''\nRelated Conversation History\nRCH_0\n'''
'''\nUser Question\nUQ_1\n'''
'''\nBot Response A\nBR_2\n'''
'''\nBot Response B\nBR_3\n'''
Please evaluate whether Bot Response is faithful to the content of
Related Conversation History to answer User Question.
Begin your evaluation by
providing a short explanation,
then you must choose the better bot response by giving
either A or B.
If the two responses are the same, you can choose NONE:
<chosen>A (or B or NONE)</chosen>.
```

Figure 12: Prompt used in GPT-4 evaluation, following Lu et al. (2023).

**Conversation History:**

[human]: Craft an intriguing opening paragraph for a fictional short story. The story should involve a character who wakes up one morning to find that they can time travel.
...(Human-Bot Dialogue Turns)... (Topic: Time-Travel Fiction)
[human]: Please describe the concept of machine learning. Could you elaborate on the differences between supervised, unsupervised, and reinforcement learning? Provide real-world examples of each.
...(Human-Bot Dialogue Turns)... (Topic: Machine learning Concepts and Types)
[human]: Discuss antitrust laws and their impact on market competition. Compare the antitrust laws in US and China along with some case studies
...(Human-Bot Dialogue Turns)... (Topic: Antitrust Laws and Market Competition)
[human]: The vertices of a triangle are at points (0, 0), (-1, 1), and (3, 3). What is the area of the triangle?
...(Human-Bot Dialogue Turns)... (Topic: Area and Circumcircle of Triangle)
...(Human-Bot Dialogue Interaction on Other Topics)...

**Question:**

What title reflects the thematic essence of the Doctor's experience with time travel as described in the story?

**Ground-Truth Answer:**

Awakening to Eternity: The Doctor's New Dawn

**Retrieved Memory (Turn-Level):**

[human]: Describe the process and write out the balanced chemical equation for the reaction that occurs when solid calcium carbonate reacts with hydrochloric acid to form aqueous calcium chloride, carbon dioxide, and water. What type of reaction is this, and what observations might indicate that the reaction is taking place?
[bot]: When solid calcium carbonate ($CaCO_3$) reacts with hydrochloric acid (HCl), the products formed are aqueous calcium chloride ($CaCl_2$), carbon dioxide ($CO_2$), and water ($H_2O$). The balanced chemical equation for this reaction is: ... (Topic: Calcium Carbonate and Hydrochloric Acid Reaction)
[human]: Write a title for your story.
[bot]: "Awakening to Eternity: The Doctor's New Dawn". (Topic: Time-Travel Fiction)
[human]: Analyze the following customer reviews from different sources for three different smartphones - the latest iPhone, Samsung Galaxy, and Google Pixel - and provide an overall rating for each phone on a scale of 1 to 10. Consider the following complex and contradictory reviews: - TechRadar's review of the latest iPhone: The new iPhone is a stunning triumph of engineering that sets a new bar for smartphone performance and camera quality. However, the incremental design and high price mean it lacks the 'wow' factor of previous iPhones. Still, its power and intelligence are unrivaled. - CNET's review of the latest Samsung Galaxy: ... Return the answer as a JSON object with the overall ratings for each phone out of 10, to one decimal place.
[bot]: "'json "iPhone": 8.5, "Samsung Galaxy": 8.0, "Google Pixel": 7.5 "'
(Topic: Smartphone Review Ratings Analysis)

**Answer (Turn-Level Retrieval):**

As an AI language model, I do not have access to the story you mentioned earlier. Please provide me with the story or the necessary details so that I can assist you better.
**(The next part of this figure 13 will be continued on the next page.)**

> **Retrieved Memory (Ours):**
>
> [human]: Craft an intriguing opening paragraph for a fictional short story. The story should involve a character who wakes up one morning to find that they can time travel.
> [bot]: In the hushed pre-dawn hours, when the world still slumbers under the cloak of night, Alex found themselves caught in the fragile boundary between sleep and wakefulness. ...
> [human]: Summarize the story with three bullet points using only nouns and adjectives, without verbs.
> [bot]: - Alex, pre-dawn awakening, time-travel discovery - Disorientation, vertigo, reality's malleability - Sunlight, new day, astonishing truth
> [human]: Is there any famous fictional character that can time travel? If so, can you rewrite your story with him or her?
> [bot]: - Doctor Who, a renowned time-traveling character from British television, is a perfect fit for this scenario. In the silent embrace of early morning, where shadows merge with the fading night, the Doctor found themselves in the thin veil separating dreams from consciousness. It was here, amidst the soft whispers of time, that an unusual sensation enveloped them—a sense of time's fabric slipping through their fingers like grains of sand. As dawn's light broke through the darkness, painting the room in hues of gold and amber, the Doctor's eyes opened to what appeared to be an ordinary day. However, upon standing, a wave of dizziness overwhelmed them, the surroundings blurred, and in that moment of confusion, the Doctor realized a profound truth: they had not only awakened to a new dawn but had also stumbled upon the incredible ability to navigate the currents of time at will.
> [human]: Write a title for your story.
> [bot]: "Awakening to Eternity: The Doctor's New Dawn" (Topic: Time-Travel Fiction)

> **Answer (Ours):**
>
> The title "Awakening to Eternity: The Doctor's New Dawn" reflects the thematic essence of the Doctor's experience with time travel as described in the story. It captures the idea of a new beginning, a sense of eternity, and the awakening of a new power that will change the course of the Doctor's life forever.

Figure 13: Case study comparing our SECOM with the *turn-level* memory baseline. When constructing the memory bank at the *turn-level*, the agent fails to provide a satisfactory response due to fragmentation: the retrieved dialogue turns can not capture a complete human-agent interaction related to the query, missing critical information highlighted in brown. The dialogue topic is emphasized in blue. Some context is omitted (replaced by ...) due to length constraints.

**Conversation History:**

[human]: Explain what's base rate fallacy and list five specific examples of ...
... (Human-Bot Dialogue Turns) ... (Topic: Base Rate Fallacy In Politics)
[human]: How many integers are in the solution of the inequality $|x + 5| < 10$?
... (Human-Bot Dialogue Turns) ... (Topic: Number of Integers in Inequalities)
[human]: How have the Alps and Rhine River influenced settlement and agriculture in Western Europe?
... (Human-Bot Dialogue Turns) ... (Topic: Alps and Rhine River Impact)
[human]: In a survey conducted at a local high school, preferences for a new school color were measured: 58% of students liked the color blue, 45% preferred green, and 22% liked both colors. If we randomly pick a student from the school, what's the probability that they would like neither blue nor green?
... (Human-Bot Dialogue Turns) ... (Topic: Probability of Liking Colors in Survey)
[human]: Imagine you are participating in a race with a group of people. If you have just overtaken the second person, what's your current position? Where is the person you just overtook?
... (Human-Bot Dialogue Turns) ... (Topic: Overtaking Positions in Race)

**Question:**

[human]: In the inequality question, how many positive integers are in the solution of the second inequality?

**Ground-Truth Answer:**

In the second inequality, $(|x + 10| < 5)$, the solution range is $(-15 < x < -5)$. Within this range, there are no positive integers, as all values of $(x)$ that satisfy the inequality are negative.

**Retrieved Memory (Session-Level):**

[human]: Explain what's base rate fallacy and list five specific examples of ...
... (Human-Bot Dialogue Turns) ... (Topic: Base Rate Fallacy In Politics)
[human]: How many integers are in the solution of the inequality $|x + 5| < 10$?
[bot]: To solve the inequality $|x + 5| < 10$, ...
[human]: What about $|x + 10| < 5$?
[bot]: To solve the inequality $|x + 10| < 5$, we ... ("overwhelmed" by other irrelevant, redundant content)
... (Human-Bot Dialogue Turns) ... (Topic: Number of Integers in Inequalities)
[human]: How have the Alps and Rhine River influenced settlement and agriculture in Western Europe?
... (Human-Bot Dialogue Turns) ... (Topic: Alps and Rhine River Impact)
[human]: In a survey conducted at a local high school, ...
... (Human-Bot Dialogue Turns) ... (Topic: Probability of Liking Colors in Survey)

**Answer (Session-Level Retrieval):**

I apologize for the confusion earlier. To answer your question, the second inequality was not provided in our previous conversation. If you could provide the inequality, I would be happy to help you find the solution and the number of positive integers in the solution set.
**(The next part of this figure 14 will be continued on the next page.)**

**Retrieved Memory (Ours):**

[human]: How many integers are in the solution of the inequality $|x + 5| < 10$?
[bot]: To solve the inequality $|x + 5| < 10$, we break it into two cases based on the definition of absolute value: 1. $x + 5 < 10$ 2. $-(x + 5) < 10$, which simplifies to $x + 5 > -10$ Solving each case: 1. $x < 5$ 2. $x > -15$ Combining these, we get $-15 < x < 5$. This inequality means that $x$ can be any number greater than $-15$ and less than $5$. To find how many integers satisfy this condition, we list them: $-14, -13, -12, -11, -10, -9, -8, -7, -6, -5, -4, -3, -2, -1, 0, 1, 2, 3, 4$. There are 19 integers that satisfy the inequality $|x + 5| < 10$.
[human]: What about $|x + 10| < 5$?
[bot]: To solve the inequality $|x + 10| < 5$, we again consider two cases: 1. $x + 10 < 5$ 2. $-(x + 10) < 5$, which simplifies to $x + 10 > -5$ Solving each case: 1. $x < -5$ 2. $x > -15$ Combining these, we get $-15 < x < -5$. This inequality means that $x$ can be any number greater than $-15$ and less than $-5$. To find how many integers satisfy this condition, we list them: $-14, -13, -12, -11, -10, -9, -8, -7, -6$. There are 9 integers that satisfy the inequality $|x + 10| < 5$.
[human]: What about $|x + 10| < 10$?
[bot]: To solve the inequality $|x + 10| < 10$, we consider two cases: 1. $x + 10 < 10$ 2. $-(x + 10) < 10$, which simplifies to $x + 10 > -10$ Solving each case: 1. $x < 0$ 2. $x > -20$ Combining these, we get $-20 < x < 0$. This inequality means that $x$ can be any number greater than $-20$ and less than $0$. To find how many integers satisfy this condition, we list them: $-19, -18, -17, -16, -15, -14, -13, -12, -11, -10, -9, -8, -7, -6, -5, -4, -3, -2, -1$. There are 19 integers that satisfy the inequality $|x + 10| < 10$. (Topic: Number of Integers in Inequalities)

**Answer (Ours):**

To find the number of positive integers that satisfy the inequality $(|x + 10| < 5)$, we need to look for positive integers in the solution set $(-15 < x < -5)$. Since there are no positive integers in this interval, the answer is 0.

Figure 14: Case study comparing our SECOM with the *session-level* memory baseline. When constructing the memory bank at the *session-level*, the agent fails to provide a satisfactory response due to the inclusion of excessive irrelevant information within a session. This distracts the generator from focusing on the truly useful information highlighted in brown. The dialogue topic is emphasized in blue. Some context is omitted (replaced by ...) due to length constraints.

**Conversation History:**

[human]: Photosynthesis is a vital process for life on Earth. Could you outline the two main stages of photosynthesis, including where they take place within the chloroplast, and the primary inputs and outputs for each stage? ... (Human-Bot Dialogue Turns)... (Topic: Photosynthetic Energy Production)

[human]: Please assume the role of an English translator, tasked with correcting and enhancing spelling and language. Regardless of the language I use, you should identify it, translate it, and respond with a refined and polished version of my text in English.
... (Human-Bot Dialogue Turns)... (Topic: Language Translation and Enhancement)

[human]: Suggest five award-winning documentary films with brief background descriptions for aspiring filmmakers to study.

[bot]: ... 5. Än Inconvenient Truth(2006) - Directed by Davis Guggenheim and featuring former United States Vice President Al Gore, this documentary aims to educate the public about global warming. It won two Academy Awards, including Best Documentary Feature. The film is notable for its straightforward yet impactful presentation of scientific data, making complex information accessible and engaging, a valuable lesson for filmmakers looking to tackle environmental or scientific subjects.
... (Human-Bot Dialogue Turns)... (Topic: Documentary Films Recommendation)

[human]: Given the following records of stock prices, extract the highest and lowest closing prices for each month in the year 2022. Return the results as a CSV string, with one line allocated for each month. Date,Open,High,Low,Close,Volume ... ... (Human-Bot Dialogue Turns)... (Topic: Stock Prices Analysis)

[human]: The city of Vega intends to build a bridge that will span the Vegona River, covering a distance of 1.8 kilometers. The proposed location falls within a seismically active area that has experienced several high-magnitude earthquakes. Given these circumstances, what would be the best approach to constructing the bridge? ... (Human-Bot Dialogue Turns)... (Topic: Seismic Bridge Construction)

**Question:**

Which documentary won two Academy Awards and what lesson does it offer to filmmakers?

**Ground-Truth Answer:**

"An Inconvenient Truth" (2006) won two Academy Awards. It offers the lesson of making complex information accessible and engaging, a valuable lesson for filmmakers looking to tackle environmental or scientific subjects.

**Memory generated by RecurSum (Wang et al., 2023):**

The dialogue provided detailed information on various topics, including the stages of photosynthesis, energy production estimates through photosynthesis for different organisms, translations and enhancements of phrases in different languages, recommendations for documentary films, a pitch for a film about overcoming adversity, analysis of stock prices, and considerations for constructing a bridge in a seismically active area. Here's a summary of the key points: ...

**Documentary Film Recommendations**: Suggested five award-winning documentary films for aspiring filmmakers to study, with brief background descriptions for each. The response included:
- "March of the Penguins" (2005) - Emperor penguins' journey in Antarctica; noted for its cinematography and portrayal of the penguins' life cycle.
- "Citizenfour" (2014) - Edward Snowden and NSA spying scandal; known for its real-time exploration of Snowden's revelations.
- "20 Feet from Stardom" (2013) - Stories of backup singers behind musical legends; celebrates unsung heroes in the music industry.
- "Man on Wire" (2008) - Philippe Petit's 1974 high-wire walk between the Twin Towers; combines personal footage, interviews, and re-enactments.
- "An Inconvenient Truth" (2006) - Al Gore's documentary on global warming; known for its presentation of scientific data.

**Answer (RSum):**

I'm sorry, but based on the updated context you provided, I don't see any previous discussion related to the question you asked. However, to answer your question, the documentary that won two Academy Awards is "Free Solo," which won for Best Documentary Feature and Best Achievement in Film Editing in 2019. ....
The film follows climber Alex Honnold as he attempts to free solo climb El Capitan in Yosemite National Park, and the filmmakers had to take risks themselves in order to capture the footage and tell the story.
**(The next part of this figure 15 will be continued on the next page.)**

---

**Retrieved Memory (Ours):**

[human]: Suggest five award-winning documentary films with brief background descriptions for aspiring filmmakers to study.

[bot]: 1. **"March of the Penguins" (2005)** - Directed by Luc Jacquet, this French feature-length nature documentary delves into the annual journey of Emperor penguins across the Antarctic. It won the Academy Award for Best Documentary Feature. The film is celebrated for its stunning cinematography and its detailed, heartfelt portrayal of the penguins' life cycle, making it a must-study for filmmakers interested in wildlife documentaries. 2. **"Citizenfour" (2014)** ... 5. **"An Inconvenient Truth" (2006)** - Directed by Davis Guggenheim and featuring former United States Vice President Al Gore, this documentary aims to educate the public about global warming. It won two Academy Awards, including Best Documentary Feature. The film is notable for its straightforward yet impactful presentation of scientific data, making complex information accessible and engaging, a valuable lesson for filmmakers looking to tackle environmental or scientific subjects.

[human]: With the spirit in the first film, craft a succinct and persuasive pitch for a film about overcoming adversity.

[bot]: Drawing inspiration from the resilience and majestic journey of Emperor penguins in "March of the Penguins," our documentary, "Unyielding Spirits: The Human Odyssey," will explore the profound stories of individuals around the globe who have faced and overcome staggering adversities. Through breathtaking cinematography and intimate storytelling, we ... (Topic: Documentary Films Recommendation)

**Answer (Ours):**

"An Inconvenient Truth" won two Academy Awards, including Best Documentary Feature. The film offers a valuable lesson to filmmakers looking to tackle environmental or scientific subjects, as it presents complex information in a straightforward yet impactful way, making it accessible and engaging to a wider audience.

Figure 15: Case study comparing our SECOM with RecurSum (Wang et al., 2023). When using RecurSum, the agent fails to provide a satisfactory answer due to information loss during the summarization process. Summarization omits essential details needed to answer the user's new request, as highlighted in brown. The dialogue topic is emphasized in blue. Some context is omitted (replaced by ...) due to length constraints.

**Conversation History:**

[human]: Photosynthesis is a vital process for life on Earth. Could you outline the two main stages of photosynthesis, including where they take place within the chloroplast, and the primary inputs and outputs for each stage? ... (Human-Bot Dialogue Turns)... (Topic: Photosynthetic Energy Production)

[human]: Please assume the role of an English translator, tasked with correcting and enhancing spelling and language. Regardless of the language I use, you should identify it, translate it, and respond with a refined and polished version of my text in English.
... (Human-Bot Dialogue Turns)... (Topic: Language Translation and Enhancement)

[human]: Suggest five award-winning documentary films with brief background descriptions for aspiring filmmakers to study.

[bot]: ... 5. Än Inconvenient Truth"(2006) - Directed by Davis Guggenheim and featuring former United States Vice President Al Gore, this documentary aims to educate the public about global warming. It won two Academy Awards, including Best Documentary Feature. The film is notable for its straightforward yet impactful presentation of scientific data, making complex information accessible and engaging, a valuable lesson for filmmakers looking to tackle environmental or scientific subjects.
... (Human-Bot Dialogue Turns)... (Topic: Documentary Films Recommendation)

[human]: Given the following records of stock prices, extract the highest and lowest closing prices for each month in the year 2022. Return the results as a CSV string, with one line allocated for each month. Date,Open,High,Low,Close,Volume ... ... (Human-Bot Dialogue Turns)... (Topic: Stock Prices Analysis)

[human]: The city of Vega intends to build a bridge that will span the Vegona River, covering a distance of 1.8 kilometers. The proposed location falls within a seismically active area that has experienced several high-magnitude earthquakes. Given these circumstances, what would be the best approach to constructing the bridge? ... (Human-Bot Dialogue Turns)... (Topic: Seismic Bridge Construction)

**Question:**

Which documentary won two Academy Awards and what lesson does it offer to filmmakers?

**Ground-Truth Answer:**

"An Inconvenient Truth" (2006) won two Academy Awards. It offers the lesson of making complex information accessible and engaging, a valuable lesson for filmmakers looking to tackle environmental or scientific subjects.

**Memory generated by ConditionMem (Yuan et al., 2023):**

- The user is seeking recommendations for award-winning documentary films, along with brief background descriptions, for educational purposes. This request likely stems from a desire to learn from successful examples in the documentary filmmaking field, possibly to improve their own filmmaking skills or to gain inspiration for their projects.
- User is interested in award-winning documentary films for educational purposes, aiming to study their backgrounds and filmmaking techniques.
- The user's request for crafting a succinct and persuasive pitch for a film about overcoming adversity is inspired by the spirit of "March of the Penguins," a documentary that beautifully captures the Emperor penguins' annual journey across the Antarctic, highlighting themes of endurance and survival against the odds. The user seeks to emulate the emotional depth and cinematic quality that made "March of the Penguins" compelling, aiming to create a pitch that resonates with similar themes of resilience and triumph over challenges.
- User is interested in creating content that focuses on themes of overcoming adversity, inspired by the emotional depth and cinematic excellence of documentaries like "March of the Penguins." They value storytelling that captures the essence of endurance and survival against the odds.
... (Other Irrelevant Memory Records) ...

**Answer (Conditional Memory):**

The documentary that won two Academy Awards is "Free Solo," which follows climber Alex Honnold as he attempts to free solo climb El Capitan in Yosemite National Park. The film offers a lesson to filmmakers on the importance of capturing authentic and compelling human stories, as well as the technical challenges of filming in extreme environments.
**(The next part of this figure 16 will be continued on the next page.)**

---

**Retrieved Memory (Ours):**

[human]: Suggest five award-winning documentary films with brief background descriptions for aspiring filmmakers to study.
[bot]: 1. **"March of the Penguins" (2005)** - Directed by Luc Jacquet, this French feature-length nature documentary delves into the annual journey of Emperor penguins across the Antarctic. It won the Academy Award for Best Documentary Feature. The film is celebrated for its stunning cinematography and its detailed, heartfelt portrayal of the penguins' life cycle, making it a must-study for filmmakers interested in wildlife documentaries. 2. **"Citizenfour" (2014)** ... 5. **"An Inconvenient Truth" (2006)** - Directed by Davis Guggenheim and featuring former United States Vice President Al Gore, this documentary aims to educate the public about global warming. It won two Academy Awards, including Best Documentary Feature. The film is notable for its straightforward yet impactful presentation of scientific data, making complex information accessible and engaging, a valuable lesson for filmmakers looking to tackle environmental or scientific subjects.
[human]: With the spirit in the first film, craft a succinct and persuasive pitch for a film about overcoming adversity.
[bot]: Drawing inspiration from the resilience and majestic journey of Emperor penguins in "March of the Penguins," our documentary, "Unyielding Spirits: The Human Odyssey," will explore the profound stories of individuals around the globe who have faced and overcome staggering adversities. Through breathtaking cinematography and intimate storytelling, we ... (Topic: Documentary Films Recommendation)

---

**Answer (Ours):**

"An Inconvenient Truth" won two Academy Awards, including Best Documentary Feature. The film offers a valuable lesson to filmmakers looking to tackle environmental or scientific subjects, as it presents complex information in a straightforward yet impactful way, making it accessible and engaging to a wider audience.

---

Figure 16: Case study comparing our SECOM with ConditionMem (Yuan et al., 2023). When using ConditionMem, the agent fails to provide a satisfactory answer due to (1) information loss during the summarization process and (2) the incorrect discarding of turns that are actually useful, as highlighted in brown. The dialogue topic is emphasized in blue. Some context is omitted (replaced by ...) due to length constraints.

