# OpenReview forum: "SeCom: On Memory Construction and Retrieval for Personalized Conversational Agents"
_ICLR.cc/2025/Conference — ICLR 2025 Poster_

### Official Review · Reviewer_wgaM · 2024-10-29

**Soundness:** 2
**Presentation:** 3
**Contribution:** 2
**Rating:** 3
**Confidence:** 5

**Summary:**

This paper constructs a memory bank with topical segments and utilizes LLMLingual-2 for denoising, thereby improving memory retrieval accuracy and reducing noise from redundancy in natural language. Experiments show that this paper outperforms turn-level and session-level baselines, as well as state-of-the-art methods. Additionally, it excels in dialogue segmentation tasks.
Further analysis and ablation studies confirm the contributions of the segment-level memory units and the compression-based denoising technique.

**Strengths:**

SeCom constructs a memory bank with topical segments and utilizes LLMLingual-2 for denoising, improving memory retrieval accuracy and reducing noise from redundancy in natural language. Experiments show that SeCom outperforms turn-level and session-level baselines and state-of-the-art methods  on LOCOMO and Long-MT-Bench+. Additionally, it excels in dialogue segmentation tasks such as DialSeg711 and TIAGE.Further analysis and ablation studies confirm the contributions of the segment-level memory units and the compression-based denoising technique.

**Weaknesses:**

- **Dialogue Scope Limitations**:  Conversational agent should encompass a broader range of dialogue types including task-oriented conversations, chit-chat, role-playing dialogues and conversational question answering. Currently, the evaluation is restricted to question-answering tasks, specifically single-turn interactions.
This narrow focus undermines the conversational agent's versatility and applicability in real-world scenarios. To provide a more comprehensive assessment, additional experimental results across various dialogue types should be included.Task-oriented datasets like MultiWOZ, open-ended conversation datasets like PersonaChat or multi-turn question answering datasets like CoQA are needed.
- **Evaluation Methodology**: The evaluation methods primarily rely on auto-machine metrics  such as bleu, roughl, bertscore and gpt4score based on large language model(LLM), which has significant limitations and may not fully capture the nuances of human interactions. The absence of human evaluations is a significant.
Incorporating human assessments such as response relevance, coherence and factual accuracy would provide a richer perspective on the conversational agent's performance. Moreover, establishing a consistency measure between human evaluations and auto-machine metrics is crucial.
Given the large language model often generate lengthy responses, some of which may be valid yet not included in the standard answers, human insights could enhance the credibility of the results.
- **Inclusion of Recent Methods**: The evaluation baselines currently lack consideration of the latest memory-enhanced methods, such as MPC and COMEDY discussed in the related work. These methods represent significant advancements in the memory with the compression of long term conversation and should be included in comparative analyses to ensure a robust evaluation framework.

**Questions:**

see Weaknesses

---

### Official Review · Reviewer_nZtP · 2024-10-29

**Soundness:** 3
**Presentation:** 4
**Contribution:** 3
**Rating:** 6
**Confidence:** 4

**Summary:**

This paper proposes the SeCom framework for memory construction and retrieval in dialogue processes. Unlike previous work that uses dialogue-level, session-level, or turn-level units to construct memory, this paper adopts segment-level, essentially topic-level, content to build memory. Additionally, a denoising operation is applied during retrieval to improve accuracy. The final experiments demonstrate that using segment-level as the memory unit effectively enhances QA results.

**Strengths:**

1. The paper adopts prefix-tuning and reflection to optimize segment prompts, further adapting them to large language models.
2. A denoising technique is used to reduce redundant information in dialogue, thereby improving retrieval accuracy.
3. The writing is well-organized, with careful and precise use of mathematical symbols.

**Weaknesses:**

1. Using segment-level content as the dialogue unit is not novel.
2. The dialogue segmentation method employed is an integration of existing methods, with no specific improvements or optimizations tailored for dialogue memory purposes.

**Questions:**

1. How does Figure 2 support the following statement? “Our findings indicate that turn-level, session-level, and summarization-based methods all exhibit limitations in terms of the accuracy of the retrieval module as well as the semantics of the retrieved content, which ultimately lead to sub-optimal responses, as depicted in Figure 1, Figure 2, and Table 1.”
2. In Figure 3(a), the formula for compression rate is given as compression rate = (#tokens after compression) / (#tokens before compression). If both “tokens after compression” and “tokens before compression” are 100, then the compression rate is 100%? Shouldn’t the compression rate be 0% in this case?
3. In Table 1, why does BM25 perform better on LOCOMO, while MPNet performs better on Long-MT-Bench+?
4. If traditional topic segmentation methods were used instead of SECOM in Table 4, what experimental results would be observed?

---

### Official Review · Reviewer_Zr2m · 2024-11-03

**Soundness:** 3
**Presentation:** 3
**Contribution:** 3
**Rating:** 6
**Confidence:** 4

**Summary:**

This paper rethinks the memory granularity for chat agents that rely on retrieval-augmented generation to memorize history information. Based on empirical insights, this paper proposes a new granularity for segmenting the cat history that lies between turn and session. In addition, the authors propose to use a prompt compression method to further compress the history before presenting it to the reader language model. Experiment results show that the proposed approach is effective in a number of long-term dialogue settings.

**Strengths:**

* The granularity of the memory is an important design choice. Similar to [1] for general RAG, this work fills in the gap in the domain of conversational agents. The conclusions established here should be impactful for future work in this domain.

* The proposal is verified in a number of settings, including various benchmarks and models. This comprehensiveness bolsters the validity of the work. I also appreciate testing the result with various evaluation metrics.

* Overall, the writing is clear. The proposal is motivated and formulated well.

[1] Dense x retrieval: What retrieval granularity should we use? Chen et al., 2023.

**Weaknesses:**

* Based on my understanding, SeCom is only tested to work with GPT-4 as the segmentation model, which is massive, costly, and thus unrealistic for many real-world applications. I would like to see a discussion on how well the framework can work without GPT-4-level models, even with some fine-tuning like in [1].

* While adding a denoising stage is an innovative proposal, there is no technical novelty in the denoising methodology itself as this work directly uses a previous prompt compression method.

* Other design aspects of the memory retrieval are ignored. For instance, it seems that this work always uses the value itself as the key, while a more advanced indexing approach could possibly reduce the need for a more “correct” value granularity [2].

[1] Dense x retrieval: What retrieval granularity should we use? Chen et al., 2023.

[2] HippoRAG: Neurobiologically Inspired Long-Term Memory for Large Language Models. Gutiérrez et al., 2024.

**Questions:**

(please also see weaknesses)

* How many segments per session do you get for the datasets you experiment on?

* Is it possible that the optimal segmentation depends on the query? For instance, round 1, 2 may be about topic A but round 2, 3 may be about topic B. Depending on whether A or B is asked, the segmentation method could be different?

* Have you conducted any human evaluation of the correctness of the segmentation? If so, what are the common error patterns?

---

### Official Review · Reviewer_Piki · 2024-11-05

**Soundness:** 3
**Presentation:** 3
**Contribution:** 3
**Rating:** 6
**Confidence:** 3

**Summary:**

The paper effectively breaks down the memory construction process into several phases: segmentation, denoising, and retrieval. This structured approach not only aids in conceptualizing the entire workflow but also provides a solid foundation for systematically enhancing each stage. SECOM further refines the process by incorporating compression-based denoising to improve memory retrieval. These adjustments reflect a well-thought-out strategy for optimizing how information is segmented, processed, and retrieved from long-term conversational memory. I believe the proposed method is both logical and persuasive.

**Strengths:**

1. The article is well-written and accessible.
2. Innovative memory granularity: It introduces segment-level memory construction, overcoming the limitations of turn-level and session-level methods, allowing for better capture of conversation structure.
3. The use of LLMLingua-2 for compression and denoising improves memory retrieval accuracy and reduces computational load.
4. The paper demonstrates SECOM’s significant advantages in memory retrieval and response quality through experiments on multiple benchmark datasets.

**Weaknesses:**

1. While the paper proposes segment-level memory construction, it lacks in-depth analysis of how to determine the optimal segment granularity, and the specific impact of different granularities on performance is not fully explored.
2. The paper does not thoroughly explore the performance of the SECOM method in dialogue systems of different scales or across different domains. Specifically, the scalability and generalization of the model in resource-constrained environments are not analyzed in detail.

**Questions:**

Please conduct additional experiments to demonstrate that the potential weaknesses do not exist.

---

> ### Author Response · Authors · 2024-11-30
> **A Kind Request for Follow-Up on Review Feedback**
>
> Dear Reviewer,
>
> Thank you once again for your time and effort in reviewing our work.
>
> We look forward to hearing your valuable feedback. If there are any clarifications or additional information needed from our side to assist with the review, please don’t hesitate to let us know.
>
> Best regards,
> The Authors of the SeCom Paper

---

### Meta-Review · Area_Chair_7N1b · 2024-12-22

**Metareview:**

This paper proposed a conversation segmentation model to segment past interactions into topical segments and retrieving compressed segments from memory during response generation, and compares this with previous methods such as ones that retrieve turns, sessions, etc. and experimented with two datasets (which is extended to others during the rebuttal period) to demonstrate the benefits of the proposed approach. Reviewers listed the novel approach that makes sense as the strength of the paper. They also asked several questions related to the  weaknesses of the work, for example, regarding the segmentation model (size) choice,  additional datasets for experimentation and the necessity of human evaluation.

**Additional Comments On Reviewer Discussion:**

Authors addressed the weaknesses listed during the rebuttal period, and provided additional experimental results and further analysis and explanations.

---

### Decision · Program_Chairs · 2025-01-22

Accept (Poster)